# Uncovering the Latent Potential of Deep Intermediate Representations

**Arnesh Batra** [1]   **Arush Gumber** [* 1]   **Aniket Khandelwal** [* 1]   **Jashn Khemani** [1]   **Anubha Gupta** [1]

## Abstract

Foundational models pretrained on huge amounts of data learn representations that evolve across depth, forming a hierarchy of embeddings with distinct semantic content and geometric structure. Contrary to the widespread practice of using only the final layer or shallow mixtures, we show that task-relevant information is distributed non-monotonically across layers and cannot be recovered by naïve aggregation. Through a geometric and empirical study across multiple modalities, we show that effective transfer depends on identifying which layers encode task-discriminative structure and how their embeddings are geometrically organized. We introduce Layer-wise Optimal Embedding Selection (LOES), a constructive spectral method that identifies task-discriminative subspaces by minimizing residual error under orthogonality and isotropy constraints. To align fine-tuning with this selection principle, we further propose Geometric Regularization (GeoReg), which enforces a simplicial structure on class manifolds and stabilizes representation geometry during fine-tuning. Across a wide range of architectures, depths, modalities, and data regimes, LOES consistently outperforms standard baselines, with gains that grow as model depth increases. Beyond accuracy, our method reveals how semantic factors are distributed across layers, thereby enabling cross-lingual and cross-modal interpretability analyses. Together, our results provide strong evidence that layerwise embedding geometry is not incidental but central to how deep models represent and transfer knowledge.

[1]SBILab, Indraprastha Institute of Information Technology Delhi, Delhi, India. Correspondence to: Arnesh Batra <arnesh23129@iiitd.ac.in>.

*Proceedings of the 43rd International Conference on Machine Learning*, Seoul, South Korea. PMLR 306, 2026. Copyright 2026 by the author(s).

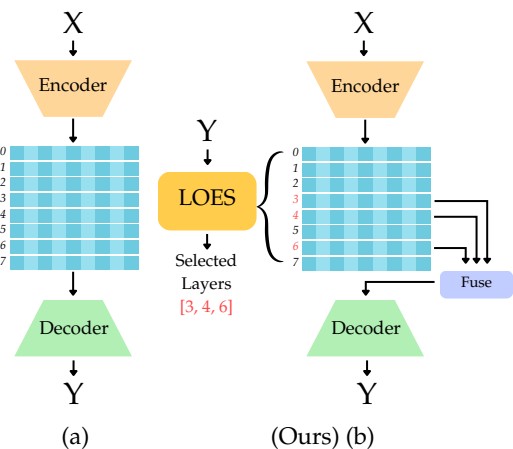

*Figure 1.* Standard vs. LOES-based transfer learning: (a) conventional transfer learning uses a single encoder layer, typically the final layer, for downstream prediction, whereas (b) LOES selects and fuses multiple task-relevant layers from the encoder hierarchy using target supervision, enabling transfer that exploits complementary information across layers.

## 1. Introduction

While foundational models (Baevski et al., 2020; Devlin et al., 2019; Girdhar et al., 2023; Oquab et al., 2024; Radford et al., 2021) define state-of-the-art transfer learning, standard adaptation protocols typically rely on the final layer output under the assumption that semantic utility increases monotonically with depth. Recent empirical analyses challenge this view, demonstrating that intermediate layers (Skean et al., 2025; Park et al., 2025; Alain & Bengio, 2017) frequently outperform the final representations by up to 10% in downstream tasks. This phenomenon occurs because the final layers often specialize overly to pretraining objectives like next-token prediction or suffer from representation collapse. In autoregressive language models specifically, a "compression valley" (de Llano et al., 2026; Li et al., 2026) emerges in mid-depth layers where the network optimally balances signal retention with noise suppression. Similar patterns are observed in vision transformers (Dosovitskiy et al., 2021), suggesting this to be a domain-general property driven by optimization dynamics rather than data modality.

To effectively recover this information, we argue that one must identify the optimal task-specific subspace formed by the top-$k$ discriminative layers regardless of their depth.

To this end, we introduce *Layer-wise Optimal Embedding Selection (LOES)*, a constructive spectral framework for subspace approximation. LOES (Figure 1) formulates layer selection as a ridge-residual optimization problem constrained by orthogonality and isotropy. By analyzing the eigenspectrum of layer-wise representations, our method systematically identifies orthogonal combinations of embeddings that maximize separability. To prevent feature collapse during the subsequent adaptation, we complement this selection with *Geometric Regularization (GeoReg)*, which is an auxiliary loss that enforces a simplicial topology on the class manifolds. Our contributions are as follows:

- We demonstrate that the optimal representations for downstream tasks are rarely the final layer alone but rather a task-specific subspace formed by a subset of intermediate layers.
- We propose *LOES*, a spectral algorithm that identifies optimal layer combinations by minimizing residual error under geometric constraints. This method outperforms learnable weighting baselines by explicitly constructing subspaces with high effective rank and low anisotropy.
- By analyzing the layer selection preferences of *LOES*, we provide a new lens for interpretability that reveals exactly where different foundational models encode specific semantic or structural attributes.
- Our analysis demonstrates that performance gains scale with model depth, and that these effects are modality- and data-agnostic working across a variety of foundational model architectures and pretraining techniques.
- We also introduce *GeoReg*, a loss function that improves stability in fine-tuning settings, mitigating feature collapse often observed in large-scale models.

## 2. Related Works

### 2.1. Suboptimality of Final Layer Embeddings

Research across modalities suggests that final-layer embeddings are not universally optimal. In LLMs, "Concept Depth" indicates that while shallow layers handle simpler tasks, deeper layers are required for complex abstractions (Jin et al., 2025). Studies have shown that mid-depth layers often encode robust features, challenging the usual emphasis on final layer representations (Lad et al., 2026; Fan et al., 2025). Similarly, in vision and multi-modal encoders, final-layer features are often suboptimal for downstream tasks when the model is trained on proxy or self-supervised objectives. Studies using image colorization as a pretraining task (Zhang et al., 2016; Zheng et al., 2025) found middle layers more effective for classification. This trend persists in autoregressive models like iGPT (Chen et al., 2020) and AIMv1 (El-Nouby et al., 2024), as well as video models like Toto (Rajasegaran et al., 2025), where intermediate layers excel in classification, tracking, and robotics.

### 2.2. Interpretability and Representational Dynamics

Researchers have introduced linear probes to monitor layer-wise classification suitability, noting monotonic increases in separability with depth (Alain & Bengio, 2017). Techniques such as SVCCA (Raghu et al., 2017) enable cross-layer comparisons, showing that networks typically converge bottom-up. BERT's (Devlin et al., 2019) hierarchy often mirrors classical NLP pipelines (de Vries et al., 2020). Recent spectral analyses identify three distinct geometric phases: warmup, entropy-seeking, and compression that define representational evolution during pretraining (Li et al., 2026). Saponati et al. (Saponati et al., 2025) present a theoretical analysis of how different pretext tasks, such as next-token prediction and masked language modeling, influence the structure of learned representations.

### 2.3. Performance Metrics for Intermediate Representations

Metrics like RankMe (Garrido et al., 2023) use effective rank as an unsupervised performance indicator, while LevyScore (Maes et al.) assesses confidence via pretraining density deviations. Layer by Layer (Skean et al., 2025) presents an information-theoretic and geometric framework which explains why mid-depth representations often outperform final layers, a property leveraged by Perception Encoder (Bolya et al., 2026) through specialized intermediate alignment.

Motivated by closed-form optimization approaches (Bertinetto et al., 2019) and recent findings suggesting that embeddings approaching an isotropic Gaussian distribution yield lower downstream prediction risk (Balestriero & LeCun, 2025), our algorithm aims to select the most effective layers for a downstream task mainly by leveraging closed-form ridge regression to predict residuals and encouraging isotropy in the resulting representations.

## 3. Methodology

We propose a unified framework that challenges the assumption of a single task-optimal layer by explicitly modeling the encoder as a hierarchy of candidate representation subspaces. Our approach consists of two coupled components: (1) a Layer-wise Optimal Embedding Selection (LOES) algorithm (Algorithm 1) that approximates the task-optimal subspace via spectral-ridge minimization, and (2) Geometric Regularization (GeoReg) as an auxiliary objective that aligns the training dynamics with the topological requirements of LOES while fine-tuning.

### 3.1. Preliminaries and Problem Formulation

Consider a foundational encoder $f_\theta(\cdot)$ that maps an input $x$ to a sequence of $L$ hierarchical representations $\mathcal{H} = \{h^{(1)}, h^{(2)}, \ldots, h^{(L)}\}$, where $h^{(\ell)} \in \mathbb{R}^{d_\ell}$. Standard transfer learning typically utilizes a projection on the final layer $\hat{y} = Wh^{(L)}$, or a learned scalar combination

$\hat{y} = W(\sum_\ell \alpha_\ell h^{(\ell)})$ (Yang et al., 2025; de Vries et al., 2020; Peters et al., 2018; Chiu et al., 2024; Alain & Bengio, 2017). Standard transfer learning restricts adaptation to the final-layer basis, assuming that downstream risk decreases monotonically with depth. This assumption is violated when task-discriminative information is distributed non-linearly across layers (Raghu et al., 2017; Kornblith et al., 2019). We therefore define the task-optimal representation as a constructive latent subspace $S \subset \mathrm{span}(\mathcal{H})$, formed by orthogonal selection of informative features across the encoder hierarchy (Saxe et al., 2013). This enables selective use of final-layer representations while integrating complementary intermediate features, yielding a geometry that minimizes empirical risk and regularizes against representation collapse (Andriopoulos et al., 2024). While we have primarily focused on classification and regression tasks along with their derivatives such as semantic segmentation, the LOES framework is inherently task-agnostic. By altering the target label space $\mathbf{Y}$, the selection mechanism can be seamlessly adapted for complex multimodal objectives like Visual Question Answering (VQA) or sequence-to-sequence generation, where informative signals are often sparsely distributed across the encoder depth.

### 3.2. Layer-wise Optimal Embedding Selection (LOES)

We propose a Layer-wise Optimal Embedding Selection (LOES) framework that works on layerwise embeddings and constructs a compact, task-discriminative subspace by iteratively selecting non-redundant layers whose embeddings explain residual task signal as well as exhibit geometric properties that promote stable probing and fine-tuning. LOES proceeds in two distinct stages. **(i) Initialization:** the first layer is selected by fitting a ridge probe of the *raw* features $\mathbf{X}_\ell$ against the original targets $\mathbf{Y}$ and minimizing a simplified score that combines fit loss with an isotropy term. **(ii) Iterative selection:** given a current selected set $S$ and cumulative prediction $\widehat{\mathbf{Y}}$, each remaining layer is orthogonalized against $\mathrm{span}(\mathbf{X}_S)$ (Eq. 3), scored against the current residual $\mathbf{R} = \mathbf{Y} - \widehat{\mathbf{Y}}$ using a composite objective (Eq. 7), and the best candidate is then refit on its *raw* features against $\mathbf{Y}$ and accumulated into $\widehat{\mathbf{Y}}$. The procedure stops when $|S| = K$.

Let $\{(\mathbf{x}_i, y_i)\}_{i=1}^N$ be a calibration set (sampled from the training set). For an encoder with $L$ layers, $\mathbf{X}_\ell \in \mathbb{R}^{N \times d_\ell}$ denotes the feature matrix for $N$ samples extracted at layer $\ell$. For classification into $C$ classes, we use one-hot encoded target values $\mathbf{Y} \in \mathbb{R}^{N \times C}$. All feature matrices are column-centered. LOES measures the *linear accessibility* of task information in a layer via closed-form Tikhonov-regularized regression (Hoerl & Kennard, 1970). Given features $\mathbf{X} \in \mathbb{R}^{N \times d}$ and targets $\mathbf{Y} \in \mathbb{R}^{N \times C}$, we consider the objective

$$\mathbf{W}^\star = \arg \min_{\mathbf{W} \in \mathbb{R}^{d \times C}} \|\mathbf{X}\mathbf{W} - \mathbf{Y}\|_F^2 + \lambda \|\mathbf{W}\|_F^2, \quad (1)$$

where $\lambda > 0$ is the Tikhonov (ridge) regularizer. This yields

a closed-form solution

$$\mathbf{W}^\star = (\mathbf{X}^\top \mathbf{X} + \lambda \mathbf{I}_d)^{-1} \mathbf{X}^\top \mathbf{Y}. \quad (2)$$

Treating classification as multi-output regression with one-hot encoded targets allows a ridge probe to produce linear class scores, with prediction given by the maximum score across classes. This formulation directly measures the linear accessibility of class information, independent of any specific classifier or loss.

Ridge regularization is essential in deep feature spaces. Encoder embeddings are often anisotropic or effectively low-rank (Li & Huang, 2026; Godey et al., 2024; Huh et al., 2023), making the unregularized normal equations ill-conditioned and leading to unstable, high-variance probes. Ridge regularization stabilizes the inversion in (2), controls the effective degrees of freedom of the fitted predictor, and yields numerically stable scores that are comparable across layers and models. When the feature dimension exceeds the calibration set size, we compute the exact solution efficiently using the Woodbury identity (Hager, 1989). Implementation details are provided in the Appendix.

To avoid selecting redundant layers, LOES evaluates candidate layers after removing components already explained by the currently selected subspace. If $S$ denotes the indices of selected layers and $\mathbf{X}_S = [\mathbf{X}_{j_1} \ \mathbf{X}_{j_2} \ \cdots] \in \mathbb{R}^{N \times D_S}$ denotes their concatenation, then for a candidate layer $\ell \notin S$ we compute ridge-regularized orthogonalized features as

$$\widetilde{\mathbf{X}}_\ell = \mathbf{X}_\ell - \mathbf{X}_S(\mathbf{X}_S^\top \mathbf{X}_S + \varepsilon \mathbf{I}_{D_S})^{-1} \mathbf{X}_S^\top \mathbf{X}_\ell. \quad (3)$$

$\widetilde{\mathbf{X}}_\ell$ is the minimum-norm residual of $\mathbf{X}_\ell$ with respect to the span of $\mathbf{X}_S$ in the ridge sense and $\varepsilon > 0$ is a small value for numerical stability. Orthogonalization ensures that the fit term in our composite score measures only the marginal explanatory power of layer $\ell$ beyond $S$; a separate redundancy term (introduced below) acts on the raw features to filter globally similar candidates before they enter the score.

Selection in LOES balances residual reduction with desired geometric properties that improve probe stability and downstream separability. We quantify three geometric diagnostics (Bihani & Rayz, 2021; Kudrjashov et al., 2024). The first is an isotropy score computed from the empirical covariance $\mathbf{\Sigma}_\ell = \frac{1}{N}\mathbf{X}_\ell^\top \mathbf{X}_\ell$. Let $\{\mu_j\}_{j=1}^{d_\ell}$ be the eigenvalues of $\mathbf{\Sigma}_\ell$ and $\overline{\mu} = \frac{1}{d_\ell} \sum_j \mu_j$. The isotropy score is

$$\mathrm{Iso}(\mathbf{X}_\ell) = \frac{\overline{\mu}}{\sqrt{\mathrm{Var}(\{\mu_j\}) + \delta}}, \quad (4)$$

with small $\delta > 0$ for numerical stability. High isotropy indicates a flat, near-uniform eigenspectrum and well-conditioned embedding. Such embeddings yield lower-variance ridge probes, more stable regression mappings, and typically smoother downstream optimization trajectories (Figure 3). The second diagnostic $\mathrm{Red}_\ell$ complements

Eq. (3) rather than duplicating it: the two mechanisms act at different stages and on different objects. Eq. (3) produces the geometric residual $\widetilde{\mathbf{X}}_\ell$ used *inside* the fit loss, whereas $\mathrm{Red}_\ell$ acts on the *raw* features through a normalized Frobenius inner product between column spaces,

$$\mathrm{Red}_\ell = \max_{j \in S} \frac{\|\mathbf{X}_\ell^\top \mathbf{X}_j\|_F}{\|\mathbf{X}_\ell\|_F \, \|\mathbf{X}_j\|_F}, \tag{5}$$

measuring global alignment with each selected layer before any projection, so that high values indicate that the candidate largely re-expresses already captured directions. In short, Eq. (3) controls *what enters* the fit loss, while Eq. (5) controls *which candidates are admitted*. The third term is an optional term only for classification, for regression and dense prediction we set it to 0. Here, on the orthogonalized features $\widetilde{\mathbf{X}}_\ell$, we compute class centroids and estimate average triangle area among random triplets of centroids,

$$\mathrm{Tri}(\widetilde{\mathbf{X}}_\ell) \approx \mathbb{E}_{(a,b,c)} \left[ \tfrac{1}{2} \sqrt{\|b-a\|^2 \|c-a\|^2 - \langle b-a, c-a \rangle^2} \right]. \tag{6}$$

This term (6) indicates that class centroids span a higher-volume simplex rather than lying near a low-dimensional line, which correlates with linear separability and robustness to perturbations. Let $\widehat{\mathbf{Y}}$ denote the cumulative ridge prediction from layers in $S$ (initialized at $\mathbf{0}$) and $\mathbf{R} = \mathbf{Y} - \widehat{\mathbf{Y}}$ the current residual. LOES fits a ridge probe on each orthogonalized candidate $(\widetilde{\mathbf{X}}_\ell, \mathbf{R})$ and computes the mean squared error loss, $\mathrm{Loss}_\ell$, which thus measures only the *marginal* contribution of layer $\ell$ beyond $S$. Candidates are scored by the composite objective

$$\mathrm{Score}(\ell) = \mathrm{Loss}_\ell + \alpha\big(1 - \mathrm{Iso}(\mathbf{X}_\ell)\big) + \gamma\,\mathrm{Red}_\ell - \eta\,\mathrm{Tri}(\widetilde{\mathbf{X}}_\ell), \tag{7}$$

with nonnegative trade-off parameters $\alpha, \gamma, \eta$. Note that Iso and $\mathrm{Red}_\ell$ are computed on the raw features $\mathbf{X}_\ell$ (intrinsic geometry, independent of $S$), while $\mathrm{Loss}_\ell$ and Tri are computed on the orthogonalized residual $\widetilde{\mathbf{X}}_\ell$ (marginal contribution beyond $S$). Upon selecting the layer $\ell^\star$ that minimizes this score, we refit a ridge probe on the *raw* features $\mathbf{X}_{\ell^\star}$ against the original targets $\mathbf{Y}$ (not against $\mathbf{R}$). The resulting prediction is accumulated to the ensemble:

$$\widehat{\mathbf{Y}} \leftarrow \widehat{\mathbf{Y}} + \mathbf{X}_{\ell^\star} \mathbf{W}_{\ell^\star},$$

after which the residual $\mathbf{R} = \mathbf{Y} - \widehat{\mathbf{Y}}$ is recomputed, the layer is appended to the selected set $S$, and the procedure is repeated until the layer budget $K$ is reached. This scoring/refit asymmetry is deliberate: scoring uses $(\widetilde{\mathbf{X}}_\ell, \mathbf{R})$ so candidates are ranked by marginal contribution, while refitting uses $(\mathbf{X}_{\ell^\star}, \mathbf{Y})$ so the downstream probe operates on unmodified encoder features at inference.

In practice LOES is applied on a modest calibration budget $N_{\mathrm{cal}} \ll$ full dataset size. The isotropy term improves conditioning of the probe and reduces variance in the fitted

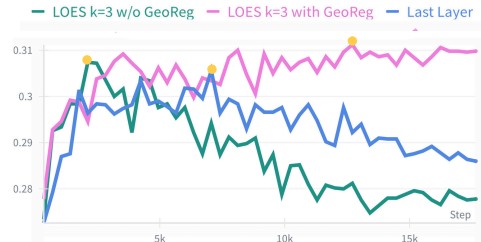

*Figure 2.* **GeoReg prevents representation collapse during fine-tuning.** Validation accuracy with trainable BERT Base (Devlin et al., 2019) on TweetEval - Emoji (Barbieri et al., 2020; 2018) dataset. Without GeoReg (green), accuracy degrades after ~10k steps despite initial gains. GeoReg (magenta) maintains stable performance. The last-layer baseline (blue) exhibits similar collapse. Dots indicate best checkpoint.

predictor, while the redundancy penalization avoids selecting layers that merely repackage existing directions. These geometric elements consistently improve stability and generalization over residual-only selection in our ablations. Furthermore, we provide a theoretical analysis (Appendix A.7) demonstrating that our isotropy maximization objective guarantees the selection of features that minimize the worst-case parameter estimation error.

### 3.3. Geometric Regularization (GeoReg)

LOES selects layers whose embeddings exhibit favorable geometry: high isotropy and well-separated class centroids. However, when the encoder is fine-tuned, gradient updates can degrade these properties, causing the fused representation to collapse into a low-dimensional subspace or class centroids to converge. This phenomenon, termed representation collapse, is well documented in self-supervised learning (Bardes et al., 2022; Balestriero & LeCun, 2025), where methods such as VICReg and LeJEPA address it through variance-covariance regularization during *pretraining*. Related geometric losses that enforce inter-class orthogonality have also been explored for metric learning (Lezama et al., 2018). GeoReg applies analogous principles at *transfer time*, preserving the geometric structure that motivated layer selection in the first place. Let $\{\mathbf{z}_i\}_{i=1}^B$ denote the fused embeddings for a minibatch, obtained by concatenating adapter (Appendix A.3) outputs from LOES-selected layers. GeoReg regularizes two complementary aspects of this representation. The first term penalizes *spectral imbalance* in the embedding covariance. Let $\boldsymbol{\Sigma} = \frac{1}{B}\sum_i (\mathbf{z}_i - \bar{\mathbf{z}})(\mathbf{z}_i - \bar{\mathbf{z}})^\top$ with eigenvalues $\{\lambda_j\}$. We define

$$\mathcal{L}_{\mathrm{iso}} = \mathrm{Var}(\{\lambda_j\}), \tag{8}$$

which is minimized when eigenvalues are uniform, corresponding to isotropic utilization of the representational capacity. The second term encourages *class separation* using the same geometric criterion as in LOES layer scoring (Eq. 6): we compute class centroids $\{\boldsymbol{\mu}_c\}$ from the fused

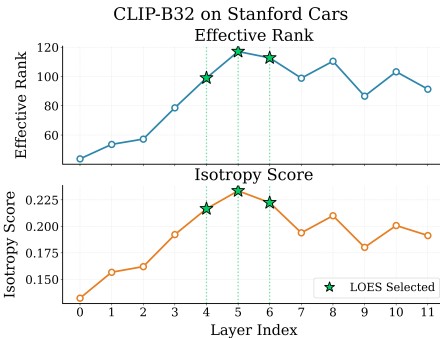

*Figure 3.* Layer-wise representation geometry for CLIP-B/32 on Stanford Cars. Effective rank (top; higher means more dimensions contribute) and isotropy score (bottom; higher means a flatter covariance eigenspectrum) peak in mid layers. Stars mark LOES-selected layers, which align with high-rank, near-isotropic representations.

embeddings within each minibatch and penalize configurations where centroids span low volume, indicating collapse toward a degenerate subspace. The combined objective is

$$\mathcal{L}_{\text{GeoReg}} = \lambda_{\text{geo}} \left( \mathcal{L}_{\text{iso}} - \log(A + \epsilon) \right), \qquad (9)$$

where $A$ denotes the centroid triangle area and $\epsilon$ is very small ensuring numerical stability of the solution.

Unlike VICReg and LeJEPA, which regularize encoder representations during pretraining, GeoReg operates on the *fused* multi-layer embeddings during downstream adaptation. This placement is deliberate: the geometry exploited by LOES must be preserved under competing fine-tuning gradients. As shown in Figure 2, without GeoReg, validation accuracy degrades after initial gains when the encoder is trainable, a signature of representational collapse. GeoReg stabilizes the optimization trajectory throughout the training. When the encoder is frozen, GeoReg has no effect since the embedding geometry remains fixed, confirming that its benefit arises specifically from constraining how fine-tuning reshapes the representation manifold.

## 4. Computational Efficiency

Table 1 shows that LOES introduces minimal computational overhead across both image and text benchmarks. In terms of time complexity, LOES adds a one-time calibration and layer-selection cost that scales linearly with the number of encoder layers and calibration samples and does not scale with training epochs. Exact time complexity and pseudocode are provided in the Appendix. During training, LOES differs from last-layer transfer only by fusing a small number of intermediate representations, while the dominant cost remains that of the encoder's forward pass, which is identical for both methods. The parameter overhead introduced by LOES arises solely from the linear probe operating on the fused representation. For a representative 100M-parameter frozen backbone, the probe adds fewer

| Dataset | Last Layer (s) | LOES $k=3$ (s) |
|---|---|---|
| *Image Classification (DINOv2-S/14)* | | |
| SUN397 | 2259 | 2307 |
| Mini-ImageNet | 1413 | 1418 |
| Stanford Cars | 459 | 473 |
| *Text Classification (ModernBERT)* | | |
| Toxic Conversations 50K | 7349 | 7353 |
| Emotion | 718 | 728 |
| MTOP Domain | 529 | 660 |

*Table 1.* End-to-end wall-clock time in seconds over 15 epochs for last-layer transfer and LOES.

than 1M parameters, corresponding to an increase of under 1% in total parameter count. Empirically, this efficiency is seen in wall-clock results, with LOES remaining within a few percent of the last-layer baseline across datasets and model depths, including ModernBERT and task-adapted variants for segmentation and regression. Overall, LOES is computationally efficient and a drop-in replacement for last-layer transfer.

## 5. Experimental setup

All experiments follow a single, consistent transfer pipeline unless otherwise stated. Layer-wise embeddings are extracted from pretrained encoders and used to train a linear probe. All probe and adapter training runs use 15 epochs. Optimization is performed with AdamW (Loshchilov & Hutter, 2019) with weight decay $1 \times 10^{-4}$ and a cosine-annealing learning-rate schedule. The probe and adapters use a learning rate of $1 \times 10^{-4}$, while backbone fine-tuning, when enabled, uses $1 \times 10^{-5}$. The random seed is fixed to 0 for all experiments. Standard batch size is 256. Layer selection and scoring are performed on a calibration split comprising 20% of the training data. Based on the calibration study (Appendix Table A3), we fix the calibration fraction to 20% in all subsequent experiments, as it consistently achieves near-peak performance across datasets with substantially fewer calibration samples.

**Hyperparameters.** LOES uses a small set of interpretable hyperparameters controlling isotropy ($\alpha$), redundancy ($\gamma$), class geometry ($\eta$), and the number of selected layers $k$. In all experiments, we use $\alpha = 1.0, \gamma = 0.5, \eta = 0.1/0.0$, and select $k \in \{3, 4\}$ layers, which consistently yields optimal or near-optimal performance. We observe that performance typically saturates after $k = 3\text{–}4$, with negligible gains beyond this range. A detailed ablation and sensitivity analysis validating these choices is provided in Appendix A.4.1 (Appendix Table A1).

**Models.** We evaluate LOES on a diverse set of pretrained encoders covering multiple pretraining paradigms. Vision models include DINOv2 (Oquab et al., 2024) and DINOv3 (Siméoni et al., 2026) (self-distillation), MAE (masked re-

| Model | Pretraining Paradigm | Selected Layers | Layer Pattern | Avg. Rel. Gain (%) |
|---|---|---|---|---|
| *Vision Encoders* | | | | |
| CLIP-B/32 | Contrastive Vision-Language | [5, 6, 4], [8, 9, 10] | Task-dependent | +7.8 |
| MAE-B/16 | Masked Image Reconstruction | [11, 10, 9] | Final layers | +44.3 |
| DINOv2-S | Self-distillation | [11, 10, 7], [11, 6, 7] | Task-dependent | +7.9 |
| DINOv3-S/16 | Self-distillation | [11, 10, 9], [11, 10, 4] | Task-dependent | +5.9 |
| DeiT-B/16 | Supervised Distillation | [8, 9, 7], [9, 10, 8] | Mid-to-late | +12.9 |
| ViT-IN21k-B/16 | Supervised Classification | [10, 9, 8], [10, 11, 9] | Late layers | +10.1 |
| *Language Encoders* | | | | |
| BERT-base (12L) | Masked Language Modeling | [11, 10, 6, 7], [7, 10, 6, 9] | Late + mid | +2.5 |
| ModernBERT (22L) | Masked Language Modeling | [3, 21, 1, 5], [1, 21, 14, 5] | Task-dependent | +17.7 |
| *Speech Encoders* | | | | |
| Wav2Vec 2.0 | Self-supervised Temporal | [3, 0, 1, 2], [7, 6, 8, 5] | Task-dependent | +33.3 |

*Table 2.* **Pretraining paradigm influences layer selection patterns and LOES gains.** Selected layers are shown for representative tasks (vision: Stanford Cars, CUB-200; language: MTOP, Emotion; speech: ASVspoof, CREMA-D). Relative gain is computed as the average percentage improvement of LOES over last-layer transfer across evaluated datasets.

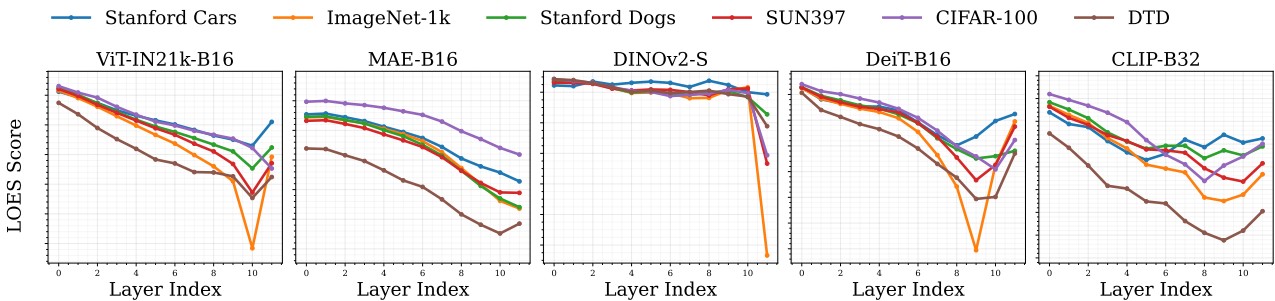

*Figure 4.* **LOES score distribution across encoder depth** (lower is better). Models pretrained exclusively on ImageNet (ViT-IN21k, MAE, DeiT) exhibit monotonically decreasing scores toward final layers, indicating task-discriminative information concentrates at depth. CLIP, pretrained on 400M diverse image-text pairs, shows comparatively flatter profiles with competitive scores in mid-depth layers, consistent with the early-to-mid layer selections reported in Table 2. These patterns suggest that pretraining data diversity influences how task-relevant information is distributed across the encoder hierarchy.

construction), DeiT (Touvron et al., 2021) and ViT (Dosovitskiy et al., 2021) (supervised or distillation-based pretraining), and CLIP (Chiu et al., 2024) (contrastive vision-language alignment). For language, we consider both classical and modern masked language models, including BERT-large (Devlin et al., 2019) and ModernBERT (Warner et al., 2025), enabling analysis of depth and large-scale pretraining effects. We additionally evaluate speech encoders such as Wav2Vec 2.0 (Baevski et al., 2020), trained via self-supervised temporal prediction.

**Tasks.** We evaluate LOES across a diverse set of downstream tasks, including classification, regression, and dense prediction. Image classification experiments are conducted on ImageNet-1K (Russakovsky et al., 2015), CUB-200 (Wah et al., 2011), Stanford Cars (Krause et al., 2013), CIFAR-100 (Krizhevsky et al., 2009), DTD (Cimpoi et al., 2014), Mini-ImageNet (Russakovsky et al., 2015), and SUN397 (Xiao et al., 2010; 2014). Text classification

is evaluated on multiple MTEB (Enevoldsen et al., 2025; Muennighoff et al., 2023) benchmarks, including Emotion (Saravia et al., 2018), Amazon Massive Intent (FitzGerald et al., 2022), Amazon Massive Scenario (FitzGerald et al., 2022), Amazon Counterfactual (O'Neill et al., 2021), MTOP Domain Classification (Li et al., 2021), Banking77 (Casanueva et al., 2020), Tweet Sentiment Extraction (Maggie, 2020), and Toxic Conversations 50K (cjadams et al., 2019). We further evaluate speech classification on ASVspoof 2019 (Wang et al., 2020), CREMA-D (Cao et al., 2014), and Google Speech Commands (Warden, 2018), regression/classification on multimodal and vision–language datasets such as Amazon Products 23 (Asaniczka, 2023), Fakeddit (Nakamura et al., 2020), FashionGen (Rostamzadeh et al., 2018) and dense prediction on semantic segmentation benchmarks Cityscapes (Cordts et al., 2016) and COCOStuff (Caesar et al., 2018). For classification, LOES is used in its standard form to select a task-discriminative

subset of layers, whose representations are concatenated and fed to a linear classifier. For regression and dense prediction, the LOES objective is adapted by replacing classification-specific terms with residual and geometric criteria suited to continuous or pixel-level supervision (see Appendix). Across all settings, the core principle remains unchanged: selecting complementary, non-redundant layers that maximize linearly accessible task signal.

## 6. Results and Discussion

### 6.1. Pretraining Paradigm Shapes Layer Selection

A central finding of our experiments is that the pretraining objective influences which layers encode task-discriminative information. The layers selected, and the extent to which LOES improves over baselines may depend on the downstream task as well. Table 2 summarizes layer selection patterns across modalities for different pretraining paradigms, while Figure 4 visualizes the LOES score distribution across layers for a given downstream task.

### 6.2. LOES Improves Existing Layer Selection Methods

LOES-selected embeddings outperform not only last-layer transfer (Figure 5, Table 3) but also learnable layer weights, fixed concatenation of final layers, and some other recently proposed selection methods.

**Vision Encoders:** Layer selection patterns correlate with pretraining data diversity (Table 2). CLIP, trained on 400 million image-text pairs, and DINOv2/DINOv3, trained on the 142-million image LVD dataset, consistently select early-to-middle layers ([5, 6, 4] for CLIP on Stanford Cars; [11, 10, 7] for DINOv2). In contrast, MAE, DeiT, and ViT-IN21k, pretrained exclusively on ImageNet variants, select predominantly final layers ([11, 10, 9] for MAE; [10, 9, 8] for ViT-IN21k). This pattern suggests that models pretrained on larger and more diverse data distribute task-relevant information more evenly across depth, whereas models trained on narrower distributions concentrate discriminative features in later layers. Consequently, explicit layer inspection via methods like LOES becomes valuable as pretraining corpora grow in scale and diversity.

On Stanford Cars with trainable DINOv2-S/14 (Table 4), last-layer transfer achieves 76.8%, last-3 concatenation achieves 79.3%, learnable weights achieve 78.0%, while LOES achieves 82.7%, demonstrating that principled selection outperforms both single-layer and naive multi-layer baselines. LOES also extends to dense prediction tasks: on Cityscapes and COCOStuff semantic segmentation (Appendix Table A13), LOES consistently selects early layers alongside the final layer ([0, 9, 11] for DINOv2; [0, 3, 11] for BEiT), with BEiT (Bao et al., 2022) showing the largest gain (+3.14 mIoU on Cityscapes). We additionally compared LOES against exhaustive search over all 220 three-layer subsets of frozen BERT-base on MTOP and Emotion (Appendix Table A5). LOES recovers a subset within 1.16

and 1.69 percentage points of the global optimum at a small fraction of the search cost, supporting its use on deeper encoders where exhaustive enumeration is infeasible.

**Language Encoders:** Model depth amplifies the benefits of layer selection. BERT-large (24 layers) improves modestly from 95.32% to 97.81% on MTOP (Table 5), with LOES selecting predominantly late layers [9, 19, 20, 21]. Modern-BERT (22 layers) improves substantially from 78.07% to 94.48%, with LOES selecting [3, 15, 1, 4], spanning early and later layers. Embeddings selected by LOES even outperform (Table 5) the use of intrinsic dimensionality of layers (Cheng et al., 2025; Razzhigaev et al., 2024) and a recent framework of layer selection (Skean et al., 2025) that introduced information-theoretic, geometric and augmentation invariance metrics like entropy and curvature.

LOES also outperforms a stronger learned fusion baseline that applies per-layer linear projections and concatenates across all 22 layers of ModernBERT-base (Appendix Table A9): LOES-4 reaches 94.19 on MTOP and 78.51 on AM-Scenario, against 93.91 and 77.51 for the all-layer learned fusion, despite using 1.7x fewer parameters and FLOPs. This indicates the gains come from the selection criterion itself rather than from the richer concatenated representation. Cross-lingual results on Amazon MASSIVE with MultilingualModernBERT-B( 22 layers) (Marone et al., 2025). (Figure 6, Table A12) reveals that LOES benefits vary substantially across languages. High-resource languages with Latin scripts (English, German, French) show moderate gains over the last-3 baseline (+2–3%). However, languages underrepresented in typical pretraining corpora show markedly larger improvements: Hindi (+6.5%), Arabic (+7.2%), and Urdu (+10.9%). Despite these differences in magnitude, LOES consistently selects layer 6 alongside the final layer across all languages, suggesting that mid-depth representations encode cross-lingually transferable structure that complements language-specific features in later layers. These results indicate that LOES may be particularly beneficial for transfer to underrepresented languages, where the final layer alone inadequately captures task-relevant information. Full cross-lingual results are provided in Appendix Table A12.

**Speech Encoders:** Wav2Vec 2.0 (Table A14) exhibits task-dependent layer selection. For spoofed speech detection (ASVspoof 2019), LOES selects early layers [3, 0, 1, 2], improving from 90.89% to 98.80%. For emotion recognition (CREMA-D), mid-depth layers [7, 6, 8, 5] are selected, improving from 37.84% to 69.06%. This heterogeneity suggests that acoustic artifacts relevant to spoofing detection manifest in early layers, while paralinguistic features for emotion recognition emerge at intermediate depths.

**Multimodal Regression Tasks:** LOES extends beyond classification to regression and multimodal settings (Appendix Table A11). On Amazon Products 23, LOES reduces RMSE from 161.80 to 154.44; on FashionGen, from 527.19 to

| Model | Method | CUB-200 | CIFAR-100 | DTD | Mini-IN | S. Dogs | S. Cars | SUN397 |
|---|---|---|---|---|---|---|---|---|
| CLIP-B32 | Last | 19.67±0.33 | 75.63±0.31 | 62.96±1.04 | 89.21±0.09 | 60.65±0.35 | 50.53±0.83 | 65.67±0.22 |
| | LOES | **20.27±0.67** | **80.59±0.24** | **70.93±0.54** | **90.93±0.07** | **61.75±0.38** | **59.95±0.56** | **72.07±0.10** |
| DINOv2-S | Last | 75.10±0.38 | 83.20±0.10 | 61.69±1.27 | 93.72±0.12 | 81.83±0.42 | 48.61±0.50 | 68.62±0.31 |
| | LOES | **82.58±0.40** | **84.31±0.04** | **68.10±1.19** | **94.27±0.09** | **84.12±0.11** | **60.86±0.28** | **71.96±0.21** |
| DINOv3-S/16 | Last | 70.86±0.69 | 84.07±0.26 | 61.99±1.56 | 93.37±0.06 | 79.69±0.35 | 77.54±0.46 | 66.69±0.10 |
| | LOES | **79.91±0.52** | **84.73±0.09** | **68.24±0.89** | **93.60±0.10** | **81.92±0.38** | **84.53±0.22** | **70.61±0.22** |
| DeiT-B/16 | Last | 65.45±0.89 | 82.49±0.16 | 58.18±1.23 | **98.63±0.05** | **93.72±0.14** | 38.88±0.65 | 61.25±0.29 |
| | LOES | **76.17±0.20** | **83.75±0.22** | **66.95±0.44** | 98.44±0.02 | 93.47±0.15 | **57.98±0.56** | **66.46±0.22** |
| MAE-B/16 | Last | 9.78±0.41 | 62.22±0.23 | 45.00±1.25 | 81.82±0.12 | 51.95±0.52 | 7.42±0.37 | 39.25±0.16 |
| | LOES | **20.39±0.44** | **68.36±0.24** | **60.51±0.68** | **84.22±0.23** | **59.93±0.30** | **15.45±0.28** | **51.41±0.22** |
| ViT-IN21k-B/16 | Last | 76.02±0.30 | 77.43±0.19 | 55.24±0.75 | **93.17±0.08** | 85.24±0.17 | 25.12±0.33 | 66.99±0.23 |
| | LOES | **81.68±0.20** | **79.44±0.16** | **63.68±0.84** | 93.15±0.10 | **85.33±0.07** | **35.56±0.12** | **69.47±0.16** |

*Table 3.* **Test-set accuracy (%) comparing the last-layer baseline and LOES with $k = 3$.** All results are reported as mean ± standard deviation over 5 independent runs with different random seeds. The best result for each model–dataset pair is highlighted in bold.

| Encoder / State | Method ($k$) | S. Cars | Mini-IN | S. Dogs | CUB-200 | DTD | CIFAR-100 | SUN397 |
|---|---|---|---|---|---|---|---|---|
| **DINOv2-S/14** **(Frozen)** | Last Layer (–) | 49.4 | 93.7 | 81.3 | 76.3 | 61.6 | 83.4 | 68.5 |
| | Learnable Weights (–) | 49.6 | 94.0 | 82.6 | 77.8 | 67.8 | 83.4 | 69.9 |
| | Last 3 Layers Concat (–) | 50.1 | 94.1 | 83.2 | 77.9 | 67.8 | 84.4 | 70.1 |
| | LOES + GeoReg (2) | 55.7 | 94.0 | 83.3 | 80.3 | 67.4 | 83.9 | 71.0 |
| | LOES + GeoReg (4) | 60.3 | 94.1 | **84.3** | 83.1 | **69.6** | **84.2** | **72.7** |
| | LOES + GeoReg (3) | 60.1 | **94.2** | 84.0 | **83.1** | 68.6 | **84.2** | 72.0 |
| **DINOv2-S/14** **(Trainable)** | Last Layer (–) | 76.8 | 93.7 | 80.5 | 79.2 | 72.6 | 90.8 | 66.4 |
| | Learnable Weights (–) | 78.0 | 93.8 | 80.2 | 80.8 | 73.4 | 91.0 | 66.8 |
| | Last 3 Layers Concat (–) | 79.3 | 93.3 | 80.4 | 80.8 | 72.4 | 90.4 | 66.9 |
| | LOES + GeoReg (2) | 80.6 | 93.9 | 81.0 | 81.2 | 73.6 | **91.1** | 68.3 |
| | LOES + GeoReg (4) | 81.9 | 94.0 | 81.1 | 81.9 | 72.8 | 91.0 | 69.0 |
| | LOES (no GeoReg) (3) | 81.9 | 94.0 | 81.2 | 82.1 | 74.7 | 91.0 | 68.9 |
| | LOES + GeoReg (3) | **82.7** | **94.8** | **81.3** | **82.2** | **74.8** | **91.1** | **69.9** |

*Table 4.* **Layer selection and geometric regularization improve transfer accuracy.** Top-1 classification accuracy (%) across seven image benchmarks using DINOv2-S/14.

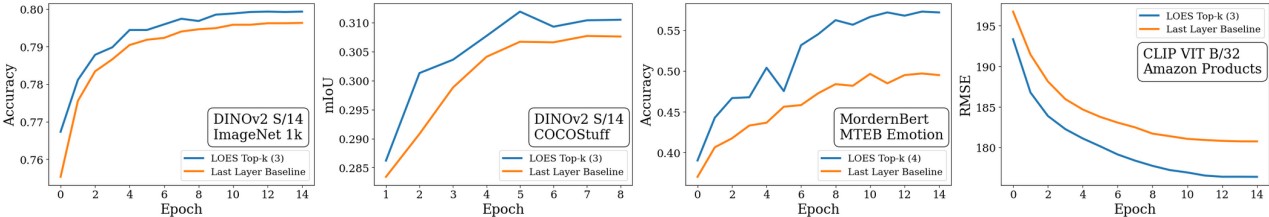

*Figure 5.* LOES boosts performance and leads to faster convergence across multiple downstream tasks like classification, segmentation and regression using popular foundation models like DINOv2, ModernBERT and CLIP.

460.46. In both cases, LOES selects mid-to-late layers ([5, 8, 10, 11] and [5, 9, 10, 11] respectively), indicating that the framework generalizes across output modalities while maintaining interpretable layer selection patterns.

## 6.3. LOES Outperforms Greedy and Random Layer Selection

We include Random baselines and a Greedy baseline (Table 6, Time means layer selection time in seconds) that picks the top 4 layers by probe accuracy (for fair comparison with LOES-4), evaluated on ModernBERT-B across two datasets. For Greedy we trained a separate probe for each layer for 2 settings: 1 or 5 epochs, incurring notable overhead ∼2.8 min for Greedy1, ∼14 min for Greedy5). LOES-4 outperforms Random (2-5), Greedy1 and matches Greedy5, while giving 32x and 160x faster layer selection than Greedy1 and Greedy5 respectively.

| Model | Mode | Emotion | AM-Intent | AM-Scenario | AM-CF | MTOP | Banking77 | Tweet-Sent | Avg. |
|---|---|---|---|---|---|---|---|---|---|
| | Last | 48.39 | 11.20 | 62.84 | 83.58 | 78.07 | 35.47 | 60.23 | 54.25 |
| | Last-4 | 50.30 | 11.94 | 63.35 | 83.88 | 79.23 | 44.67 | 60.52 | 56.27 |
| **MBERT-B** | Entropy-4 | 53.78 | 13.79 | 74.55 | 83.88 | 91.91 | 51.33 | 62.27 | 61.07 |
| | Curvature-4 | 50.60 | 11.13 | 61.43 | 83.88 | 76.68 | 44.25 | 61.83 | 55.68 |
| | Intrinsic-4 | 54.13 | 13.92 | 73.71 | 82.99 | 90.36 | 47.92 | 65.03 | 61.15 |
| | LOES-4 | **56.85** | **14.96** | **78.51** | **85.07** | **94.48** | **58.45** | **65.94** | **64.89** |
| | Last | 59.37 | 15.37 | 82.15 | 90.75 | 95.32 | 62.54 | 67.36 | 67.55 |
| | Last-4 | 62.89 | 16.57 | 85.00 | 91.04 | 96.55 | 76.65 | 68.09 | 70.11 |
| **BERT-L** | Entropy-4 | 63.34 | 17.28 | 85.10 | 91.34 | 96.76 | 75.74 | **69.75** | 71.33 |
| | Curvature-4 | 66.06 | 17.11 | **87.99** | 84.78 | 97.60 | 82.11 | 68.83 | 72.07 |
| | Intrinsic-4 | 66.39 | 15.60 | 85.27 | 87.46 | **98.29** | 82.11 | 68.03 | 71.88 |
| | LOES-4 | **67.00** | **17.45** | 87.46 | **91.94** | 97.81 | **82.50** | 69.14 | **73.33** |

*Table 5.* **Test accuracy (%) across MTEB classification datasets.** Average is computed over all datasets excluding Toxic. Horizontal rules separate *Last*, *Last-4*, and *LOES* ($k = 4$). Best result per model block is bolded (MBERT- represents ModernBERT).

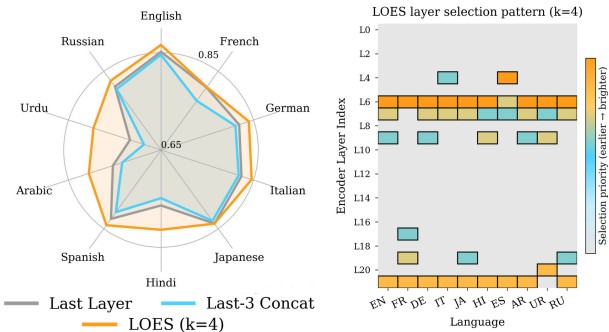

*Figure 6.* **Cross-lingual evaluation on Amazon Massive Scenario (mmBERT-B).** *Left:* LOES ($k$=4) outperforms baselines, with larger gains on underrepresented languages (Hindi +6.5%, Arabic +7.2%, Urdu +10.9% over last-3). *Right:* LOES consistently selects mid-depth layer 6 alongside the final layer across languages, indicating cross-lingually transferable structure at intermediate depths.

| Model | Mode | $k$ | AM-Scenario | MTOP | Time (s) |
|---|---|---|---|---|---|
| ModernBERT-B | Last | 1 | 62.84 | 78.07 | – |
| ModernBERT-B | Random | 2 | 69.26 | 89.07 | – |
| ModernBERT-B | Random | 3 | 66.78 | 88.57 | – |
| ModernBERT-B | Random | 4 | 77.47 | 93.88 | – |
| ModernBERT-B | Random | 5 | 76.80 | 93.22 | – |
| ModernBERT-B | Greedy1 | 4 | 76.76 | 91.06 | 168 |
| ModernBERT-B | Greedy5 | 4 | **78.81** | **94.55** | 840 |
| ModernBERT-B | LOES | 4 | 78.51 | 94.48 | 5.22 |

*Table 6.* Comparison of LOES with Random-$k$ and Greedy layer selection on MBERT-B. LOES-4 outperforms Random and Greedy1, while Greedy5 achieves comparable performance but at significantly higher selection time.

## 7. Conclusion

We presented LOES, a spectral framework for identifying task-discriminative layers in pretrained encoders. Our experiments reveal that task-relevant information is distributed non-monotonically across depth, challenging the widespread reliance on final-layer representations. The distribution of useful features depends systematically on pretraining: models trained on large, diverse corpora encode transferable structure in earlier layers, while those trained on narrower distributions concentrate information at depth. This finding suggests that as foundation models scale and diversify, principled layer selection becomes increasingly valuable. LOES consistently outperforms last-layer transfer, learnable weighting, and prior selection methods across vision, language, speech, and multimodal benchmarks. Cross-lingual evaluation reveals that underrepresented languages benefit disproportionately, suggesting practical value for low-resource transfer. Beyond accuracy improvements, LOES provides interpretable insights into how pretrained models organize knowledge. The layer selection patterns expose which depths encode task-relevant structure, offering a complementary view to existing probing methodologies. We hope this work encourages further investigation into the geometry of intermediate representations and their role in effective transfer learning.

## Software and Data Availability

Code and configuration files for reproducing the experiments are available at https://github.com/arnesh2212/loes.

## Accessibility

We have aimed to make the paper accessible by using standard LaTeX typesetting, vector-based figures where possible, descriptive captions, and visualizations that do not rely solely on color for interpretation.

## Acknowledgements

The authors thank the Infosys Centre for AI and Indraprastha Institute of Information Technology Delhi (IIIT-Delhi) for financial support and for providing computational infrastructure used in this work. We also thank the members of SBILab for helpful discussions and feedback.

## Impact Statement

This paper presents work whose goal is to improve transfer learning and interpretability for pretrained foundation models. By identifying task-relevant intermediate layers, LOES can reduce reliance on full-model adaptation and provide insight into how useful information is distributed across model depth. This may make downstream adaptation more efficient and more transparent in some settings. We do not introduce new datasets involving sensitive personal information, and all experiments are conducted on established public benchmarks.

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

# A. Appendix

## A.1. Discussion and Future Directions

The results in this work consistently show that only a small subset of intermediate representations exhibit favorable geometric and statistical properties, particularly with respect to isotropy, reduced redundancy, and stable ridge-based linear alignment. These observations suggest several concrete and feasible directions for future research that directly build on the principles validated in our experiments.

**Parameter-Efficient Adaptation.** Our empirical findings indicate that layers selected based on residual alignment and isotropy tend to contribute complementary information for downstream tasks. This naturally aligns with parameter-efficient fine-tuning methods such as low-rank adaptation and lightweight adapters, where adaptation capacity must be carefully allocated. A promising direction is to use the same layer-wise diagnostics to determine where adaptation modules should be inserted, focusing updates on layers that are both linearly accessible and well-conditioned. Such integration may be particularly effective in low-data regimes, where our results already suggest improved robustness through regularization and spectral balance.

**Unsupervised and Noisy Regimes.** Several components emphasized in this work, particularly isotropy and redundancy, do not depend on label information and therefore remain meaningful in unsupervised settings. This suggests a feasible extension in which intermediate representations are selected based on intrinsic geometric properties alone, providing a principled way to identify stable and complementary features prior to downstream use. In settings with limited or noisy supervision, our results indicate that ridge regularization and spectral balance jointly reduce sensitivity to spurious correlations. Future work could explore adaptive strategies that emphasize geometry-driven criteria when labels are unreliable, while gradually incorporating residual alignment as supervision improves. Such approaches may be especially relevant for real-world data, where clean labels are often scarce or imperfect.

**Scaling, Multimodality, and Generation.** As models grow deeper and are applied to more diverse tasks, exhaustive layer-wise analysis becomes increasingly costly. However, the core operations used here, ridge regression, orthogonal projection, and covariance-based isotropy estimation, admit efficient approximations. Hierarchical or block-level selection strategies could preserve the selection behavior observed in our experiments while remaining practical for large-scale architectures. The same principles extend naturally to multimodal and generative settings, where redundancy across modalities or conditioning features is common. Explicitly favoring isotropic and orthogonal representations may help identify features that contribute stable and complementary information prior to fusion or conditioning.

**Interpretability and Robustness.** The layer-wise scores introduced in this work provide a structured view of how representation geometry evolves across depth and tasks. Analyzing trends in isotropy, redundancy, and residual alignment can offer insight into why certain layers consistently support downstream adaptation better than others. This geometry-driven perspective complements existing interpretability approaches by grounding explanations in measurable spectral properties. Moreover, because isotropy and redundancy do not rely on labels, these ideas are well suited to unsupervised, low-data, and noisy settings. Emphasizing geometry-driven criteria in such regimes may reduce sensitivity to unreliable supervision, while ridge-based residual fitting provides a controlled mechanism for incorporating task information when available.

Overall, these directions highlight how the empirical patterns and theoretical insights identified in this work can be extended to larger models, broader tasks, and more challenging data regimes, while remaining grounded in the same underlying principles.

## A.2. Limitations

Theoretical and modeling assumptions limit guarantees. Our analysis and the isotropy objective are derived under simplified priors (e.g., a rotationally invariant task prior) that are useful as conservative, worst-case safeguards but do not exactly match every real downstream problem. Tasks with strong, structured priors or domain-specific signal can favor particular spectral directions, so forcing isotropy is not guaranteed to be strictly optimal in every case. Consequently, the diagnostics and theorems should be read as guidance about structural risk and probe stability rather than as absolute causal claims about which features are "true" for a given task.

Hyperparameters and calibration sensitivity remain a practical limitation. LOES depends on a few interpretable knobs (ridge strength, isotropy weight, redundancy weight, class-geometry weight, and the number of selected layers) and on a small calibration set. Selection and final performance can vary when these settings are poorly chosen or when the calibration data is non-representative of the deployment distribution. Crucially, however, our empirical results show that the method is robust: even under suboptimal hyperparameter settings or modestly misspecified training conditions, LOES typically improves over the single-layer baseline. Ridge regularization and the isotropy penalty in particular reduce variance and

make the probe less brittle to noisy labels or small calibration budgets, so practical gains often persist without extensive tuning. A second practical caveat is that models pretrained narrowly on a single distribution (e.g. supervised ImageNet distillation in DeiT-B/16) already concentrate task-discriminative signal in the final layer, so the headroom for LOES is small and gains can be marginal or slightly negative (see Appendix A7).

Computation and greedy search introduce additional practical constraints. The greedy selection procedure is efficient and effective in our experiments but is not guaranteed to find the global optimum; early choices can affect later residuals. Likewise, exact closed-form solves and eigendecompositions scale poorly as embedding dimension or spatial resolution grows. In practice these costs are mitigated by small calibration budgets, low layer budgets (we use $k \in 3, 4$), and approximate linear-algebra techniques when necessary, and the net effect is still a reproducible accuracy gain across modalities. Nevertheless, users should be aware that extreme model sizes, strong distribution shift in calibration data, or highly noisy labels may reduce the margin of improvement and may require lightweight approximations or modest validation to retain the benefits described in the paper.

### A.3. Layer Fusion Mechanisms

A.3.1. ADAPTORS

Let

$$\mathcal{S} = \{l_1, l_2, \ldots, l_k\}$$

denote the set of layer indices selected by LOES, where $k = |\mathcal{S}|$. For an input sample $x$, the hidden representation extracted from layer $l_i$ is denoted as

$$\mathbf{h}_{l_i} \in \mathbb{R}^D$$

where $D$ is the hidden dimensionality of the foundation model. To reduce dimensionality and learn task-specific transformations, each selected layer representation is passed through an independent lightweight adapter module. Each adapter consists of a Layer Normalization layer, followed by a linear projection and a GELU activation:

$$\mathbf{z}_{l_i} = \text{GELU}\left(W_i \, \text{LN}(\mathbf{h}_{l_i}) + b_i\right)$$

where

$$W_i \in \mathbb{R}^{d_p \times D}, \qquad b_i \in \mathbb{R}^{d_p}, d_p = 256,$$

The transformed representations from all selected layers are then concatenated to form a fused representation:

$$\mathbf{z}_{\text{fused}} = [\mathbf{z}_{l_1}; \mathbf{z}_{l_2}; \ldots; \mathbf{z}_{l_k}] \in \mathbb{R}^{kd_p}$$

A.3.2. PROBES

The fused representation is subsequently passed through a task-specific prediction head consisting of Layer Normalization, Dropout, and a linear transformation:

$$\hat{\mathbf{y}} = W_c \, \text{Dropout}\left(\text{LN}(\mathbf{z}_{\text{fused}})\right) + b_c$$

where

$$W_c \in \mathbb{R}^{C \times d_{\text{fused}}}$$

and $C$ is the number of output classes for classification tasks. For regression settings, $C = 1$.

The overall architecture enables LOES-selected intermediate representations to be efficiently fused while maintaining a lightweight parameter footprint through low-dimensional adapters.

### A.4. Additional Results

A.4.1. EFFECT OF HYPERPARAMETERS

LOES selects a subset of $K$ layers by minimizing a composite objective that balances residual task fitting and geometric regularization. At each selection step, the score for a candidate layer is

$$\mathcal{L} = \mathcal{L}_{\text{res}} + \alpha \left(1 - \text{Iso}\right) + \gamma \, \text{Red} - \eta \, \text{Tri}, \tag{10}$$

where $\mathcal{L}_{\text{res}}$ denotes the residual regression loss, Iso measures representation isotropy, Red captures similarity with previously selected layers, and Tri encourages class-separating geometry via centroid-based triangular area.

| Method | $\alpha$ | $\gamma$ | $\eta$ | $K$ | $n_{\text{cal}}$ | Test Acc ↑ |
|---|---|---|---|---|---|---|
| Last Layer | – | – | – | 1 | – | 81.37 |
| Residual Only | – | – | – | 4 | 64 | 90.61 |
| Residual Only | – | – | – | 4 | 512 | 90.93 |
| LOES | 0.0 | 0.0 | 0.0 | 4 | 64 | 90.61 |
| LOES | 0.0 | 0.0 | 0.0 | 4 | 512 | 90.93 |
| LOES | 0.0 | 0.0 | 0.01 | 4 | 512 | 91.06 |
| LOES | 0.0 | 0.1 | 0.0 | 4 | 512 | 90.97 |
| LOES | 0.0 | 0.5 | 0.0 | 4 | 512 | 93.00 |
| LOES | 0.5 | 0.0 | 0.0 | 4 | 64 | 95.90 |
| LOES | 1.0 | 0.5 | 0.1 | 4 | 64 | 95.90 |
| LOES | 1.0 | 0.5 | 0.1 | 4 | 512 | **95.96** |

*Table A1.* **Ablation of LOES hyperparameters on MTOP Domain Classification.** All results are reported using **ModernBERT-B**. $\alpha$ controls isotropy regularization, $\gamma$ penalizes redundancy with previously selected layers, and $\eta$ promotes class-separating geometry. Higher accuracy is better.

| Method | $\alpha$ | $\gamma$ | $\eta$ | $K$ | $n_{\text{cal}}$ | Selected Layers | Test Acc ↑ |
|---|---|---|---|---|---|---|---|
| Last Layer | – | – | – | 1 | – | [12] | 90.89 |
| Last-4 Concat | – | – | – | 4 | – | [9,10,11,12] | 94.83 |
| LOES | 0.0 | 0.0 | 0.0 | 4 | 512 | [8, 11, 12, 10] | 95.53 |
| LOES | 0.0 | 0.1 | 0.0 | 4 | 512 | [8, 0, 1, 2] | 97.90 |
| LOES | 0.0 | 0.5 | 0.0 | 4 | 512 | [8, 0, 1, 2] | 97.90 |
| LOES | 1.0 | 0.5 | 0.1 | 4 | 64 | [1, 0, 3, 2] | 98.13 |
| LOES | 1.0 | 0.5 | 0.1 | 4 | 512 | [1, 0, 3, 2] | **98.30** |

*Table A2.* **Cross-modal hyperparameter validation on ASVspoof 2019 LA with Wav2Vec 2.0 (frozen).** The default configuration ($\alpha$=1.0, $\gamma$=0.5, $\eta$=0.1) selected for text in Table A1 also yields the best performance in the audio modality, with under 0.3 percentage-point spread across all non-zero settings. Even the weakest LOES configuration improves over Last-4 Concat and the last-layer baseline.

### A.4.2. COMPARISON AGAINST GLOBALLY OPTIMAL SUBSETS

LOES uses a greedy selection rule, which is not guaranteed to recover the globally optimal subset of $K$ layers. To quantify the gap, we exhaustively trained and evaluated all $\binom{12}{3}$=220 three-layer subsets of frozen BERT-base on two MTEB benchmarks and compared each against the greedy LOES choice.

LOES recovers a subset within 1.16 percentage points (MTOP) and 1.69 percentage points (Emotion) of the globally optimal one, sharing two of three layers with it on both datasets. Crucially, the exhaustive search required training all 220 subsets, costing several GPU-hours even on BERT-base. For deeper encoders the search space grows rapidly: ModernBERT-base (22 layers) gives $\binom{22}{3}$=1,540 subsets, and BERT-large (24 layers) gives $\binom{24}{3}$=2,024. LOES, in contrast, requires a small calibration set and a few closed-form computations followed by a single training run, and yields the largest gains precisely on these deeper models (Table 5).

**Isotropy weight $\alpha$.** Non-zero isotropy regularization is essential for strong performance. Setting $\alpha = 0$ leads to noticeably lower accuracy, while moderate values yield substantial gains. On MTOP Domain Classification, the best-performing configurations consistently occur at $\alpha = 0.5$ to $\alpha = 1.0$, indicating that mildly encouraging isotropic representations improves linear separability without suppressing task-relevant signal.

**Redundancy weight $\gamma$.** Contrary to acting purely as a penalty, the redundancy term can positively contribute to performance. Increasing $\gamma$ from zero improves accuracy in several settings, with the best results achieved at $\gamma = 0.5$. This suggests that a moderate penalty works best: very large $\gamma$ over-penalizes useful overlap between layers, while $\gamma = 0$ admits redundant layers that merely repackage already-selected directions.

**Geometric weight $\eta$.** The triangular class-geometry term has a stabilizing effect but does not require precise tuning. Performance remains stable across a broad range of $\eta \in [0, 0.2]$, once suitable layers have been selected. This indicates that class-geometry regularization serves as a secondary refinement rather than a dominant driver of layer selection.

**Number of selected layers $K$ and calibration size $n_{\text{cal}}$.** All experiments use $K$=4. As shown in Appendix Table A4, accuracy continues to improve for some encoders beyond this point but at a roughly linear cost in head parameters and

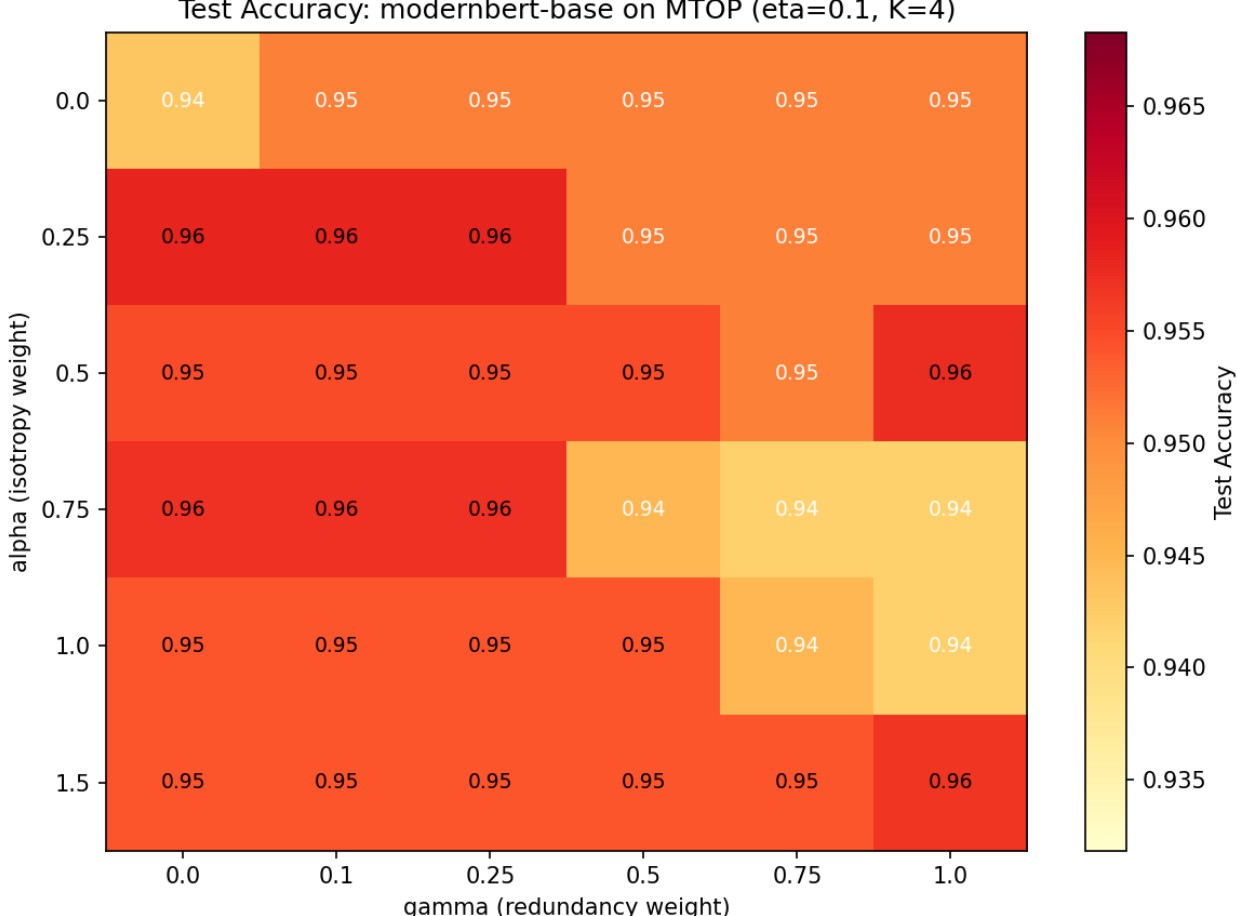

*Figure 7.* **2D sensitivity sweep over $\alpha$ and $\gamma$ on MTOP (ModernBERT-B, $K$=4).** The accuracy surface is a broad plateau: the default ($\alpha$=1.0, $\gamma$=0.5) reaches 95.90%, within 0.25 percentage points of the grid optimum (96.15%), and even the weakest configuration in the grid (94.80%) outperforms the last-layer baseline (81.37%) by more than 13 points. LOES is therefore insensitive to fine hyperparameter tuning within a wide regime.

fused-representation dimension, so $K \in \{3, 4\}$ offers the best accuracy-efficiency tradeoff. Calibration is sample-efficient: $n_{\text{cal}}$=64 consistently matches or closely approaches the best performance, demonstrating that LOES is robust to calibration set size.

**Summary.** The optimal regime

$$\alpha = [0.5, 1.0], \quad \gamma = 0.5, \quad \eta \in [0, 0.2], \quad K = 4$$

balances residual task fitting with representation geometry. This regime is stable across modalities: the same default configuration is near-optimal for text (Table A1) and for audio (Table A2), and the 2D $\alpha$–$\gamma$ surface in Figure 7 forms a broad plateau rather than a sharp optimum. These findings validate the design of LOES and support its use without per-task hyperparameter tuning.

### A.4.3. NON-LINEAR PROBING
We did some additional experiments using a ReLU activation in the probe (A.3):

$$\hat{\mathbf{y}} = \text{ReLU}\left(W_c \, \text{Dropout}\left(\text{LN}(\mathbf{z}_{\text{fused}})\right) + b_c\right)$$

Table A6 shows the results on ModernBERT-B for AM-Scenario and MTOP datasets.

| Cal (%) | Method | Emotion | AM-Intent | AM-Scenario | AM-CF | MTOP | Banking77 | Tweet-Sent | Avg. |
|---|---|---|---|---|---|---|---|---|---|
| – | Last | 48.39 | 11.20 | 62.84 | 83.58 | 78.07 | 35.47 | 60.23 | 54.25 |
| – | Last-4 | 50.30 | 11.94 | 63.35 | 83.88 | 79.23 | 44.67 | 60.52 | 56.27 |
| 0.01 | LOES | 54.43 | 14.83 | 78.45 | 85.67 | 94.03 | 58.88 | 66.20 | 64.07 |
| | *Selected* | *[5,15,14,20]* | *[3,15,1,4]* | *[3,15,1,4]* | *[14,6,15,21]* | *[1,15,3,21]* | *[1,15,4,5]* | *[1,15,4,20]* | – |
| 0.05 | LOES | 53.63 | 15.03 | 75.89 | 83.58 | 92.16 | 57.77 | 66.32 | 63.48 |
| | *Selected* | *[4,15,20,21]* | *[3,15,1,4]* | *[3,15,4,5]* | *[6,4,3,10]* | *[3,15,4,21]* | *[4,15,5,1]* | *[1,15,4,21]* | – |
| 0.10 | LOES | 55.94 | 15.00 | 78.65 | 84.48 | 94.73 | 58.55 | 65.97 | 64.76 |
| | *Selected* | *[1,15,14,20]* | *[3,15,1,2]* | *[3,15,1,4]* | *[1,6,19,21]* | *[1,15,0,4]* | *[1,16,4,5]* | *[1,15,4,6]* | – |
| 0.20 | LOES | 55.09 | 14.59 | 79.12 | 84.78 | 93.98 | 57.67 | 65.97 | 64.46 |
| | *Selected* | *[1,15,14,5]* | *[3,15,1,11]* | *[3,15,1,2]* | *[6,4,19,21]* | *[1,15,3,4]* | *[1,15,4,5]* | *[1,15,4,5]* | – |
| 0.30 | LOES | 56.45 | 14.79 | 78.78 | 86.27 | 94.14 | 58.49 | 66.11 | 65.00 |
| | *Selected* | *[1,15,14,5]* | *[3,15,1,2]* | *[3,15,1,2]* | *[6,4,3,21]* | *[1,15,3,4]* | *[1,15,4,5]* | *[1,15,4,5]* | – |
| 0.40 | LOES | 57.10 | 14.86 | 79.35 | 85.07 | 94.05 | 57.77 | 66.17 | 65.48 |
| | *Selected* | *[1,15,14,5]* | *[3,15,1,2]* | *[3,16,1,2]* | *[6,4,3,19]* | *[1,15,3,4]* | *[1,15,4,5]* | *[1,15,4,5]* | – |
| 0.50 | LOES | 56.60 | 14.39 | 79.02 | 87.46 | 94.41 | 56.86 | 66.64 | 65.62 |
| | *Selected* | *[1,15,14,5]* | *[3,15,1,2]* | *[3,15,1,2]* | *[1,6,3,10]* | *[1,15,3,4]* | *[1,15,4,5]* | *[1,15,4,5]* | – |

*Table A3.* **Comparison of Last, Last-4, and LOES ($k = 4$) on MTEB classification tasks.** We report test accuracy (%) using ModernBERT-B. *Last* uses the final encoder layer, while *Last-4* concatenates the final four layers. *LOES ($k = 4$)* selects four layers using a calibration set comprising 1–50% of the training data and concatenates their representations. For each calibration fraction, the first row reports accuracy and the italicized row lists the selected encoder layers per dataset. Average accuracy is computed across all seven datasets.

### A.4.4. SENSITIVITY TO THE RIDGE REGULARIZER $\lambda$

Throughout all experiments, the Tikhonov regularizer $\lambda$ in the ridge probe (Eq. 1) is fixed at $10^{-3}$ for numerical stability and is not tuned per dataset. To verify that LOES is not sensitive to this choice, we sweep $\lambda$ across four orders of magnitude on two MTEB classification tasks using ModernBERT-base. As shown in Appendix Table A10, LOES outperforms the Last-4 concatenation baseline by 14–16 percentage points across all values of $\lambda$, with less than $1.4\%$ variation across the sweep range. This confirms that the ridge regularizer functions purely as a conditioning safeguard for the closed-form solve, not as a performance-critical hyperparameter, and that the default value $\lambda = 10^{-3}$ can be used without per-dataset tuning.

### A.4.5. COMPARISON WITH VARIANCE-COVARIANCE REGULARIZERS

GeoReg is designed to preserve spectral isotropy and class-centroid separation during fine-tuning, where competing task gradients can otherwise erode both. This concern is shared with self-supervised regularizers such as VICReg (Bardes et al., 2022) and SIGReg (Balestriero & LeCun, 2025), which similarly target representation collapse via variance-covariance penalties. The key distinction is timing: VICReg and related methods regularize the encoder during *pretraining*, whereas GeoReg operates on the *fused* multi-layer representation during downstream adaptation, where the LOES-selected geometry must be defended against task-specific gradient updates.

To validate this design choice, we compare GeoReg against a VICReg-style adaptation applied at the same fusion point on TweetEval-Emoji using BERT-base with $K = 3$. Without geometric regularization, validation accuracy degrades after approximately 10k steps due to representation collapse, consistent with the trajectory shown in Figure 2. The VICReg-adapted baseline achieves $30.69\%$, marginally above the last-layer baseline at $30.15\%$. GeoReg reaches $31.04\%$ and exhibits stronger resistance to late-training collapse, indicating that the combination of spectral isotropy and centroid-volume terms is more effective than variance-covariance penalties alone in the transfer setting. A more detailed comparison covering VICReg, SIGReg, and additional regularizers is in preparation for the main paper but omitted here due to page constraints.

### A.4.6. T-SNE FOR LOES SELECTED LAYERS

LOES-selected layers produce tighter, more separated clusters compared to last-layer and last-K baselines. Figure 8 shows that the adapted representations (after training) of LOES have better cluster compactness and separability than the last layer(s) (also adapted), consistent with the gains reported earlier.

### A.5. Statistical Evaluation Protocol

We conduct a comprehensive statistical analysis to assess whether LOES provides consistent and significant improvements over a last-layer baseline. All experiments are performed using five independent random seeds per model and dataset to

| Model | Mode | $k$ | Accuracy (%) | Selected Layers |
|-------|------|-----|-------------|-----------------|
| **ModernBERT-B** | Last | 1 | 81.37 | [21] |
| | Last-3 | 3 | 82.44 | [19,20,21] |
| | LOES | 1 | 92.50 | [3] |
| | LOES | 2 | 95.51 | [1,21] |
| | LOES | 3 | 95.99 | [3,21,1] |
| | LOES | 4 | 95.76 | [3,21,1,4] |
| | LOES | 5 | **96.01** | [1,21,3,4,10] |
| | LOES | 6 | 95.92 | [1,20,3,4,10,21] |
| | LOES | 7 | 95.78 | [1,21,3,4,10,17,19] |
| | LOES | 8 | 95.67 | [3,21,1,4,10,19,20,18] |
| | LOES | 9 | 95.12 | [1,21,3,4,15,20,18,19,16] |
| | LOES | 10 | 95.42 | [3,21,1,4,11,20,19,18,17,15] |
| **BERT-base** | Last | 1 | 96.67 | [11] |
| | Last-3 | 3 | 97.40 | [9,10,11] |
| | LOES | 1 | 96.08 | [7] |
| | LOES | 2 | 97.15 | [11,10] |
| | LOES | 3 | 97.97 | [11,10,6] |
| | LOES | 4 | 97.74 | [11,10,6,7] |
| | LOES | 5 | 97.83 | [6,10,7,11,9] |
| | LOES | 6 | 97.74 | [11,10,6,9,7,8] |
| | LOES | 7 | 98.18 | [7,10,6,9,8,11,4] |
| | LOES | 8 | 98.34 | [11,10,9,6,7,8,4,5] |
| | LOES | 9 | 98.38 | [11,10,6,9,7,8,4,5,3] |
| | LOES | 10 | **98.43** | [7,10,9,6,11,8,5,4,3,1] |

*Table A4.* **Effect of LOES layer count ($k$) on MTOP domain classification.** Test accuracy (%) reported for varying $k$. Horizontal rules separate *Last*, *Last-3*, and LOES sweeps. Best result per model is in bold.

| Dataset | Method | Layers | Test Acc |
|---------|--------|--------|----------|
| MTEB MTOP | Global Optimal | [3, 6, 11] | **97.83** |
| | LOES | [11, 10, 6] | 97.67 |
| | Last Layer | [11] | 95.51 |
| MTEB Emotion | Global Optimal | [3, 10, 11] | **71.08** |
| | LOES | [11, 10, 9] | 69.39 |
| | Last Layer | [11] | 68.40 |

*Table A5.* **LOES against exhaustive search over $\binom{12}{3}$=220 subsets** on frozen BERT-base. LOES reaches within 1.16 percentage points of the optimum on MTOP and 1.69 on Emotion, sharing 2 of 3 selected layers with the optimal subset in both cases. Note that probes were trained for 10 epochs for these experiments

account for variability arising from initialization and data ordering.

**Mean and Variance Estimation.** For each model and dataset pair, we compute the mean and standard deviation of validation and test accuracies across seeds. These statistics are reported as mean $\pm$ standard deviation in all result tables and serve as stable estimates of expected performance and variability.

**Paired Significance Testing.** To evaluate whether LOES significantly outperforms the baseline, we perform paired two-sided $t$-tests for each model and dataset combination. The paired design matches LOES and baseline runs using identical random seeds, thereby controlling for seed-specific noise.

Let $d_i$ denote the difference in accuracy between LOES and the baseline for seed $i$. The test statistic is computed as

$$t = \frac{\bar{d}}{s_d/\sqrt{n}},$$

| Model | Mode (Non-Linear Probe) | AM-Scenario | MTOP |
|---|---|---|---|
| ModernBERT-B | Last | 62.40 | 80.62 |
| ModernBERT-B | Last-4 | 64.40 | 81.00 |
| ModernBERT-B | LOES-4 | **77.41** | **94.04** |

*Table A6.* Performance of LOES under non-linear probing (ReLU) on ModernBERT-B. LOES generalizes effectively beyond linear probes, significantly outperforming standard baselines.

| **Model** | **Dataset** | $n$ | $t$-**stat** | $p$-**value** | **Significant** |
|---|---|---|---|---|---|
| CLIP-B32 | CUB-200 | 5 | 1.69 | 1.67e-01 | No |
| CLIP-B32 | CIFAR-100 | 5 | 26.05 | 1.29e-05 | Yes |
| CLIP-B32 | DTD | 5 | 19.68 | 3.93e-05 | Yes |
| CLIP-B32 | Mini-IN | 5 | 25.54 | 1.40e-05 | Yes |
| CLIP-B32 | S. Dogs | 5 | 3.60 | 2.32e-02 | Yes |
| CLIP-B32 | S. Cars | 5 | 46.72 | 1.26e-06 | Yes |
| CLIP-B32 | SUN397 | 5 | 80.83 | 1.40e-07 | Yes |
| DINOv2-S | CUB-200 | 5 | 26.64 | 1.18e-05 | Yes |
| DINOv2-S | CIFAR-100 | 5 | 18.84 | 4.67e-05 | Yes |
| DINOv2-S | DTD | 5 | 13.46 | 1.76e-04 | Yes |
| DINOv2-S | Mini-IN | 5 | 13.67 | 1.66e-04 | Yes |
| DINOv2-S | S. Dogs | 5 | 12.03 | 2.74e-04 | Yes |
| DINOv2-S | S. Cars | 5 | 53.10 | 7.53e-07 | Yes |
| DINOv2-S | SUN397 | 5 | 38.81 | 2.63e-06 | Yes |
| DINOv3-S/16 | CUB-200 | 5 | 17.66 | 6.03e-05 | Yes |
| DINOv3-S/16 | CIFAR-100 | 5 | 4.54 | 1.05e-02 | Yes |
| DINOv3-S/16 | DTD | 5 | 6.40 | 3.05e-03 | Yes |
| DINOv3-S/16 | Mini-IN | 5 | 3.65 | 2.17e-02 | Yes |
| DINOv3-S/16 | S. Dogs | 5 | 7.20 | 1.97e-03 | Yes |
| DINOv3-S/16 | S. Cars | 5 | 30.98 | 6.47e-06 | Yes |
| DINOv3-S/16 | SUN397 | 5 | 33.72 | 4.61e-06 | Yes |
| DeiT-B/16 | CUB-200 | 5 | 23.79 | 1.85e-05 | Yes |
| DeiT-B/16 | CIFAR-100 | 5 | 7.73 | 1.50e-03 | Yes |
| DeiT-B/16 | DTD | 5 | 15.81 | 9.35e-05 | Yes |
| DeiT-B/16 | Mini-IN | 5 | -7.96 | 1.35e-03 | Yes |
| DeiT-B/16 | S. Dogs | 5 | -9.38 | 7.19e-04 | Yes |
| DeiT-B/16 | S. Cars | 5 | 95.94 | 7.08e-08 | Yes |
| DeiT-B/16 | SUN397 | 5 | 31.71 | 5.90e-06 | Yes |
| MAE-B/16 | CUB-200 | 5 | 31.90 | 5.76e-06 | Yes |
| MAE-B/16 | CIFAR-100 | 5 | 38.74 | 2.65e-06 | Yes |
| MAE-B/16 | DTD | 5 | 18.53 | 4.99e-05 | Yes |
| MAE-B/16 | Mini-IN | 5 | 18.52 | 5.01e-05 | Yes |
| MAE-B/16 | S. Dogs | 5 | 36.68 | 3.30e-06 | Yes |
| MAE-B/16 | S. Cars | 5 | 32.26 | 5.51e-06 | Yes |
| MAE-B/16 | SUN397 | 5 | 168.03 | 7.53e-09 | Yes |
| ViT-IN21k-B/16 | CUB-200 | 5 | 30.62 | 6.78e-06 | Yes |
| ViT-IN21k-B/16 | CIFAR-100 | 5 | 23.07 | 2.09e-05 | Yes |
| ViT-IN21k-B/16 | DTD | 5 | 19.10 | 4.43e-05 | Yes |
| ViT-IN21k-B/16 | Mini-IN | 5 | -0.52 | 6.31e-01 | No |
| ViT-IN21k-B/16 | S. Dogs | 5 | 1.41 | 2.33e-01 | No |
| ViT-IN21k-B/16 | S. Cars | 5 | 54.10 | 6.99e-07 | Yes |
| ViT-IN21k-B/16 | SUN397 | 5 | 15.09 | 1.12e-04 | Yes |

*Table A7.* **Statistical significance of LOES vs last-layer baseline on the test set.** Paired $t$-tests are conducted over $n$=5 seeds.

where $\bar{d}$ is the mean of the per-seed differences, $s_d$ is their standard deviation, and $n = 5$ is the number of seeds.
We apply this test independently to validation and test accuracies. A result is considered statistically significant if the corresponding $p$-value is below $0.05$. While validation results are analyzed for completeness, we primarily report test-set significance, as it reflects generalization performance.

| Dataset | # Models | $F$-statistic | $p$-value | Significant |
|---|---|---|---|---|
| CUB-200 | 6 | 24983.42 | 9.15e-44 | Yes |
| CIFAR-100 | 6 | 5813.34 | 3.60e-36 | Yes |
| DTD | 6 | 107.39 | 1.29e-15 | Yes |
| Mini-IN | 6 | 7710.42 | 1.22e-37 | Yes |
| S. Dogs | 6 | 14983.21 | 4.22e-41 | Yes |
| S. Cars | 6 | 20108.81 | 1.24e-42 | Yes |
| SUN397 | 6 | 8300.11 | 5.04e-38 | Yes |

*Table A8.* **One-way ANOVA across models using LOES test accuracy.** The analysis tests whether different pretrained encoders achieve significantly different performance on each dataset. All datasets show statistically significant differences across models ($p < 0.05$).

| Method | MTOP | AM-Scenario |
|---|---|---|
| Last Layer | 80.96 | 63.45 |
| Learnable Concat (all 22 layers) | 93.91 | 77.51 |
| LOES-4 | **94.19** | **78.51** |

*Table A9.* **LOES against a stronger learned fusion baseline on ModernBERT-base.** Learnable Concat applies a per-layer linear projection to a low-dimensional space and concatenates across all 22 layers, giving it 1.7x more head parameters and FLOPs than LOES-4. LOES-4 still matches or exceeds this baseline on both tasks, indicating that the gains come from the selection criterion rather than from richer downstream fusion.

**Interpretation of Results.** The paired $t$-tests indicate that LOES yields statistically significant improvements over the last-layer baseline in the majority of model and dataset combinations. Gains are particularly pronounced on fine-grained benchmarks such as CUB-200, Stanford Cars, and DTD. In a small number of cases, differences are not statistically significant, indicating that LOES does not degrade performance when improvements are absent.

The sign of the $t$-statistic further reflects the direction of change. Positive values correspond to LOES outperforming the baseline, while negative values indicate marginal baseline advantages, which are rare and often not statistically significant.

**Characterizing Failure Cases.** Two model and dataset pairs show a statistically significant negative effect: DeiT-B/16 on Mini-ImageNet and DeiT-B/16 on Stanford Dogs. The absolute differences are small (0.19% and 0.25% respectively), so the regression is mild in practice but consistent across seeds. DeiT-B/16 is pretrained via supervised distillation exclusively on ImageNet variants, which concentrates task-discriminative information in the final layers. In this setting the last layer is already close to optimal, intermediate layers contribute noise rather than complementary signal, and LOES has little room to improve over the baseline. This pattern is consistent with the broader trend in Table 2: narrowly pretrained models (DeiT, MAE, ViT-IN21k) concentrate discriminative features in late layers, whereas models pretrained on diverse corpora (CLIP, DINOv2) distribute them across depth and benefit most from LOES.

**Cross-Model Analysis.** To examine whether different pretrained encoders exhibit significantly different performance under LOES, we additionally perform one-way analysis of variance tests across models for each dataset. The results show that model choice remains a significant factor in performance, confirming that LOES complements rather than replaces the inductive biases introduced by pretraining.

Overall, the statistical analysis demonstrates that LOES provides robust and reproducible improvements across architectures and datasets. The observed gains are consistent across random seeds and are supported by rigorous paired significance testing, providing strong evidence for the effectiveness of the proposed method.

### A.6. Additional Discussion

This subsection provides additional discussion and supporting empirical results that complement the main analysis and further validate the proposed framework across different settings, metrics, and evaluation protocols.

**Effect of Layer Count on Performance.** Appendix Table A4 reports performance as a function of the number of selected layers $k$ on MTOP. The trend depends on the encoder: ModernBERT-base peaks at $k=5$ (96.01%) with marginal differences across $k \in \{3, 4, 5\}$, while BERT-base improves nearly monotonically from $k=1$ (96.08%) to $k=10$ (98.43%). However, larger $k$ comes at a roughly linear cost in fused-representation dimensionality and head parameters: for ModernBERT, moving from $k=3$ to $k=5$ adds 67% more parameters in the head for a 0.02 point gain; for BERT-base, moving from $k=4$ to $k=10$ adds 2.5x parameters for a 0.69 point gain. We therefore use $k \in \{3, 4\}$ throughout the paper as the best accuracy-efficiency tradeoff rather than as the saturation point of the underlying curve.

| Method / $\lambda$ | MTOP | AM-Scenario |
|---|---|---|
| Last-4 Concat | 79.23 | 63.35 |
| LOES, $\lambda = 10^{-5}$ | 94.80 | 79.02 |
| LOES, $\lambda = 10^{-4}$ | 94.27 | 79.62 |
| LOES, $\lambda = 10^{-3}$ | 94.45 | 78.45 |
| LOES, $\lambda = 10^{-2}$ | 94.73 | 79.76 |

*Table A10.* **Sensitivity of LOES to the ridge regularizer** $\lambda$. Test accuracy (%) on MTOP and AM-Scenario using ModernBERT-base with $k = 4$. LOES is robust across four orders of magnitude in $\lambda$, with all configurations outperforming the Last-4 baseline by a wide margin.

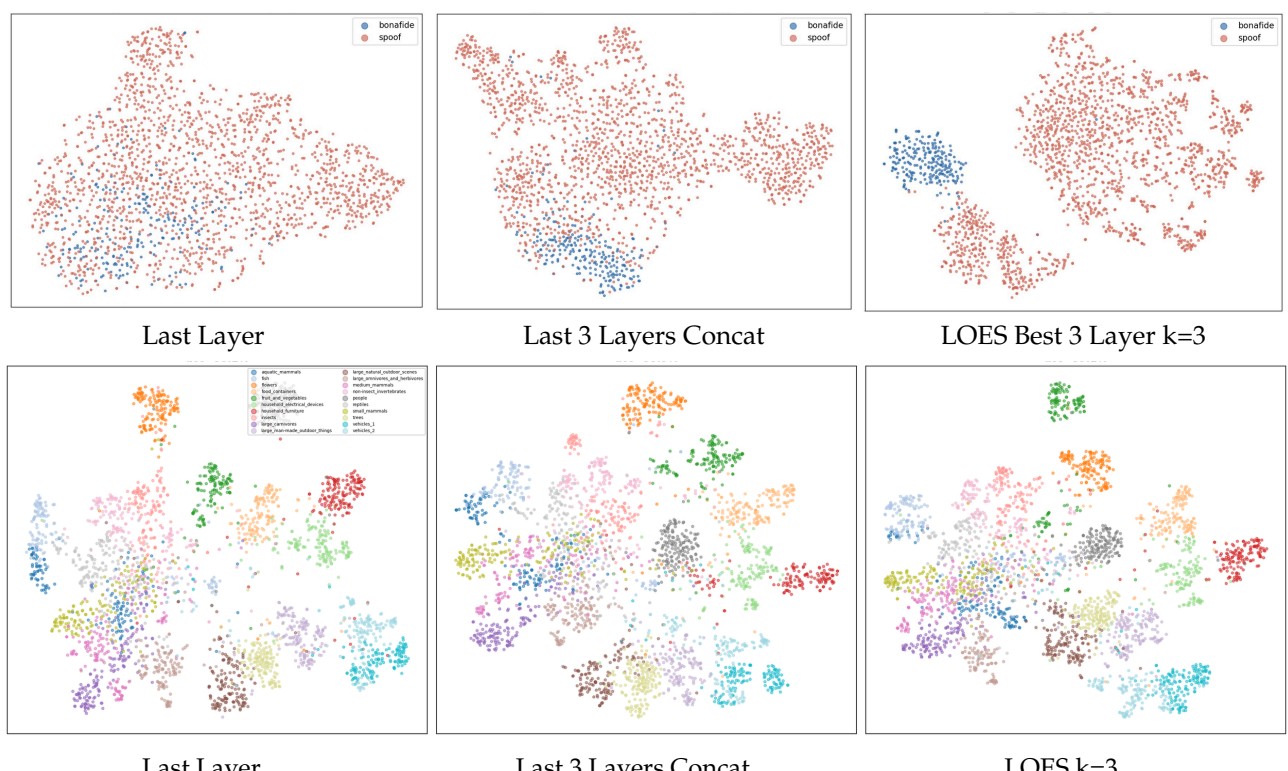

*Figure 8.* t-SNE visualizations comparing standard last-layer representations with LOES-selected layer fusion on CIFAR-100 (top, using DINOv2) and ASVspoof 2019 (bottom, using Wav2Vec 2.0). On CIFAR-100, simple concatenation of the last three layers exhibits moderate class mixing, whereas LOES ($k = 3$; layers 6, 7, and last) produces tighter and better-separated clusters, demonstrating the advantage of selective layer fusion. On ASVspoof 2019, last-layer embeddings show strong overlap between bonafide and spoof samples, while concatenation of the last three layers provides only marginal improvement. In contrast, LOES-selected representations achieve substantially clearer separation, with LOES Best Layer ($k = 1$) identifying a more informative representation and LOES Best 3 Layers ($k = 3$) yielding the most compact and well-separated clusters.

**Calibration-Free and Minimal-Calibration Baselines.** Appendix Table A3 includes Last and Last-4 baselines that require no calibration data. These results show that naive concatenation of final layers provides limited improvements, whereas LOES consistently yields stronger performance even with small calibration budgets.

**Statistical Robustness Across Random Seeds.** Appendix Table A7 reports paired $t$-test results comparing LOES with the last-layer baseline across multiple vision benchmarks. The results indicate that the observed improvements are statistically significant across random seeds and not driven by favorable initialization.

**Cross-Model Variability Under LOES.** Appendix Table A8 presents one-way ANOVA results assessing performance differences across pretrained encoders under LOES. While LOES improves accuracy across models, encoder choice remains statistically significant, indicating that LOES preserves and leverages model-specific inductive biases rather than homogenizing representations.

| Dataset | Task | Method | Test Metric | Layers Selected |
|---------|------|--------|-------------|-----------------|
| Amazon Products 23 | Regression | Last | RMSE ↓ 161.80 | 12 |
| | | Last 3 | RMSE ↓ 156.05 | 10,11,12 |
| | | Learnable | RMSE ↓ 159.23 | – |
| | | LOES (2) | RMSE ↓ 158.14 | 9,10 |
| | | LOES (3) | RMSE ↓ 156.37 | 8,9,10 |
| | | LOES (4) | **RMSE ↓ 154.44** | 5,8,10,11 |
| FashionGen | Regression | Last | RMSE ↓ 527.19 | 12 |
| | | Last 3 | RMSE ↓ 646.30 | 10,11,12 |
| | | Learnable | RMSE ↓ 637.31 | – |
| | | LOES (2) | RMSE ↓ 486.87 | 9,10 |
| | | LOES (3) | RMSE ↓ 481.65 | 9,10,11 |
| | | LOES (4) | **RMSE ↓ 460.46** | 5,9,10,11 |
| Fakeddit | Classification | Last | Acc ↑ 0.8790 | 12 |
| | | Last 3 | Acc ↑ 0.8859 | 10,11,12 |
| | | Learnable | **Acc ↑ 0.8971** | – |
| | | LOES (2) | Acc ↑ 0.8822 | 0,12 |
| | | LOES (3) | Acc ↑ 0.8852 | 0,11,12 |
| | | LOES (4) | Acc ↑ 0.8865 | 0,10,11,12 |

*Table A11.* **Layer selection results using CLIP ViT-B (base) with joint image–text embeddings.** Results are reported on Amazon Products 23 and FashionGen (least squares regression; lower RMSE is better) and Fakeddit (binary classification; higher accuracy is better). Baseline methods and LOES variants are separated by horizontal rules.

| Language | Last | Last-3 | LOES ($k$=4) | $\Delta$ vs Last-3 | Selected Layers |
|----------|------|--------|--------------|--------------------|-----------------|
| English (EN) | 85.2 | 84.6 | 86.6 | +2.0 | [6, 21, 7, 9] |
| French (FR) | 80.9 | 77.5 | 81.0 | +3.5 | [6, 21, 7, 19] |
| German (DE) | 81.9 | 81.1 | 84.0 | +2.9 | [6, 21, 7, 9] |
| Italian (IT) | 82.4 | 81.8 | 84.6 | +2.8 | [6, 21, 7, 4] |
| Japanese (JA) | 83.7 | 83.0 | 83.7 | +0.7 | [6, 21, 7, 19] |
| Spanish (ES) | 82.5 | 80.8 | 84.1 | +3.3 | [4, 21, 6, 7] |
| Russian (RU) | 81.1 | 80.5 | 82.6 | +2.1 | [6, 21, 7, 19] |
| Hindi (HI) | 76.4 | 74.9 | 81.4 | +6.5 | [6, 21, 9, 7] |
| Arabic (AR) | 75.4 | 73.4 | 80.6 | +7.2 | [6, 21, 7, 9] |
| Urdu (UR) | 71.7 | 68.7 | 79.6 | +10.9 | [6, 20, 9, 7] |

*Table A12.* **Cross-lingual LOES results on Amazon Massive Scenario using mmBERT-B (22 layers).** Accuracy (%) reported for last-layer, last-3 concatenation, and LOES with $k$=4. $\Delta$ vs Last-3 denotes the improvement of LOES over last-3 concatenation in percentage points. Languages below the mid-rule (Hindi, Arabic, Urdu) are underrepresented in typical pretraining corpora and show substantially larger LOES gains. Layer indices are zero-indexed; layer 22 is the final layer.

**Additional Multimodal and Regression Results.** Appendix Table A11 reports layer-selection results for multimodal least squares regression and binary classification tasks using CLIP ViT-B. LOES improves performance across heterogeneous output spaces, including settings evaluated with RMSE, demonstrating that the method extends beyond classification accuracy.

**Extended Segmentation and Speech Results.** Appendix Tables A13 and A14 provide additional evidence that LOES generalizes to dense prediction and speech classification tasks.

### A.6.1. ADAPTING LOES FOR SEMANTIC SEGMENTATION

The standard LOES framework operates on image-level (pooled) embeddings for classification tasks. To extend LOES to dense prediction tasks such as semantic segmentation, we adapt the calibration and selection procedure to operate at the pixel level while preserving the core algorithmic structure.

**Pixel-Level Calibration.** For a calibration set of $N_{\text{cal}}$ images, we extract intermediate representations from all encoder layers and reshape them to spatial feature maps. Let $\mathbf{F}_\ell \in \mathbb{R}^{B \times H_f \times W_f \times d}$ denote the spatial features at layer $\ell$, where $H_f = W_f = H_{\text{img}}/p$ is the feature resolution and $p$ is the patch size. We downsample the ground-truth segmentation masks

| Dataset | Model | Last Layer | LOES ($k$=3) | $\Delta$ | Selected Layers |
|---------|-------|-----------|-------------|----------|-----------------|
| Cityscapes | DINOv2-S | 52.80 | 53.63 | +0.83 | [0, 9, 11] |
| | DINOv3-S/16 | 52.04 | 53.01 | +0.97 | [5, 9, 11] |
| | BEiT-B/16 | 36.07 | 39.21 | +3.14 | [0, 3, 11] |
| COCOStuff | DINOv2-S | 30.77 | 31.19 | +0.42 | [0, 8, 11] |
| | DINOv3-S/16 | 31.15 | 31.16 | +0.01 | [5, 9, 11] |
| | BEiT-B/16 | 16.71 | 17.83 | +1.12 | [0, 3, 11] |

*Table A13.* **LOES extends to semantic segmentation.** Mean IoU (%) reported for last-layer and LOES with $k$=3 on Cityscapes and COCOStuff-164k. $\Delta$ denotes improvement in percentage points. All models use frozen encoders with a linear segmentation head. Cityscapes models were trained for 20 epochs; COCOStuff models were trained for 8 epochs due to larger dataset size. Layer indices are zero-indexed; layer 11 is the final layer.

to match this resolution using nearest-neighbor interpolation.

To construct a tractable calibration set, we randomly sample $M$ pixels per image from valid (non-ignored) spatial locations. This yields a pixel-level feature matrix $\mathbf{X}_\ell \in \mathbb{R}^{N \times d}$, where $N = N_{\text{cal}} \times M$, and corresponding one-hot encoded class labels $\mathbf{Y} \in \mathbb{R}^{N \times C}$.

**Layer Scoring and Selection.** Given pixel-level embeddings, the LOES scoring and selection procedure remains unchanged. For each candidate layer $\ell$, we compute the closed-form ridge regression solution as in Eq. (2) and evaluate the composite score:

$$\text{Score}(\ell) = \mathcal{L}_{\text{ridge}}(\mathbf{X}_\ell, \mathbf{Y}) + \alpha\big(1 - \text{Iso}(\mathbf{X}_\ell)\big) + \gamma \, \text{Red}_\ell, \tag{11}$$

where $\text{Iso}(\cdot)$ is the isotropy score (Eq. (4)) and $\text{Red}_\ell$ penalizes redundancy with previously selected layers (Eq. (5)). We omit the triangular class-geometry term ($\eta = 0$) for segmentation, as computing class centroids over hundreds of pixel classes is computationally prohibitive and the term provides marginal benefit in this setting.

Layer selection proceeds greedily as in Algorithm 1: we first select the layer minimizing the initial score, then iteratively add layers that best reduce the residual prediction error while avoiding redundancy with the current selection.

**Decoder Architecture.** Selected layer features are fused via per-layer linear adapters followed by concatenation, consistent with the classification pipeline. The fused representation is passed through a lightweight convolutional segmentation head and bilinearly upsampled to the original image resolution. The encoder remains frozen throughout training; only the adapters and segmentation head are optimized.

**Summary.** The key adaptation for segmentation is the shift from image-level to pixel-level calibration: instead of pooling spatial tokens, we treat each valid pixel as an independent sample for ridge regression and isotropy computation. This preserves the geometric and residual-based selection principles of LOES while accommodating dense prediction tasks. As shown in Appendix Table A13, this adaptation consistently improves mIoU over last-layer baselines across multiple encoders and datasets.

| Model | Method | ASVspoof 2019 | CREMA-D | Google Speech Commands |
|-------|--------|--------------|---------|------------------------|
| Wav2Vec 2.0 (Frozen) | Last Layer | 90.89 | 37.84 | 85.15 |
| | LOES (2) | **98.80** | 68.97 | 92.41 |
| | LOES (3) | 98.70 | 69.01 | **92.60** |
| | LOES (4) | 98.74 | **69.06** | 92.51 |

*Table A14.* **Layer selection results using Wav2Vec 2.0 with a frozen encoder.** Classification accuracy is reported on ASVspoof 2019, CREMA-D, and Google Speech Commands. For LOES with $k = 4$, selected layer indices are shown in the order: ASVspoof 2019 / CREMA-D / Google Speech Commands.

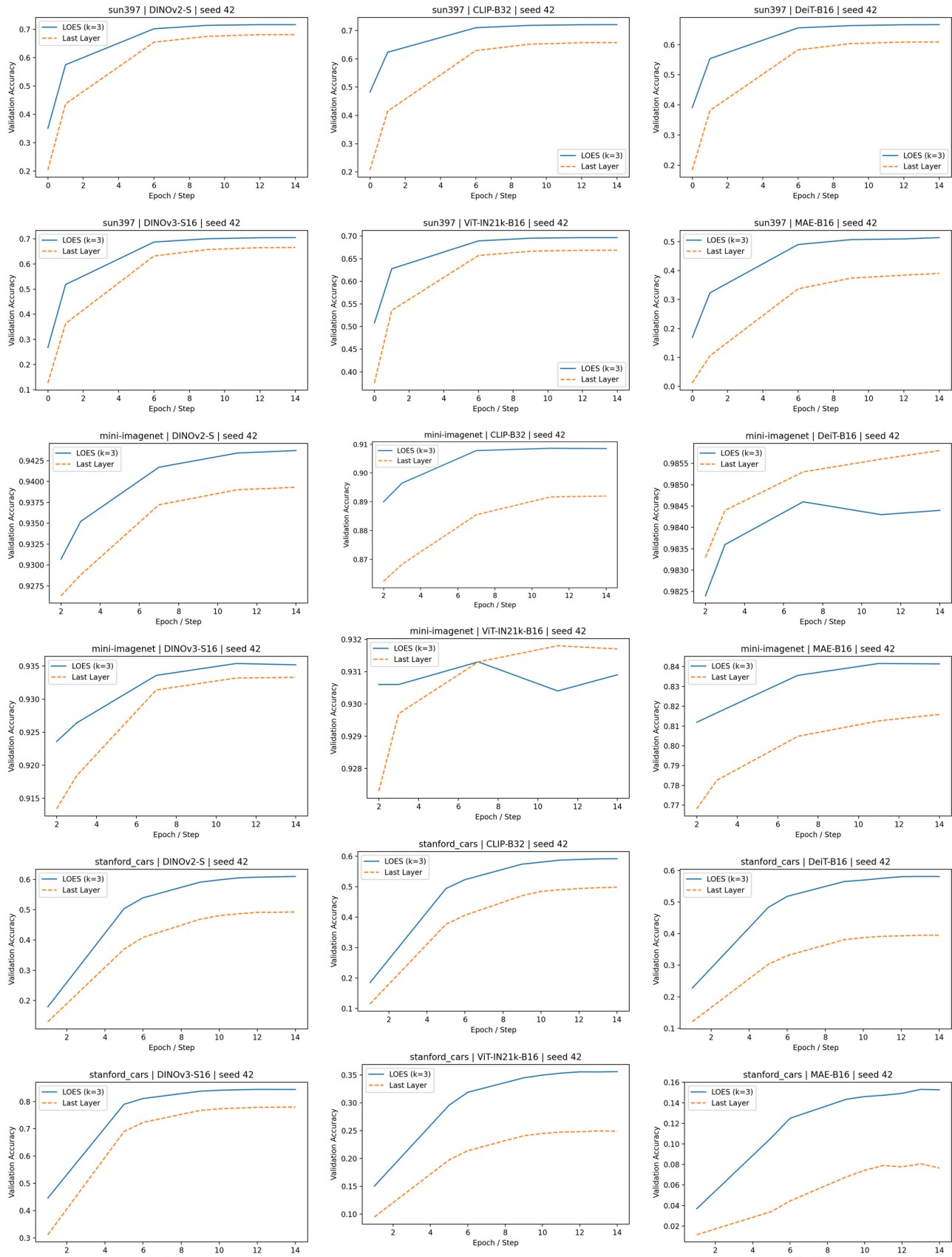

*Figure 9.* Epoch-wise validation accuracy comparing LOES (k=3) and last-layer baselines across datasets and models.

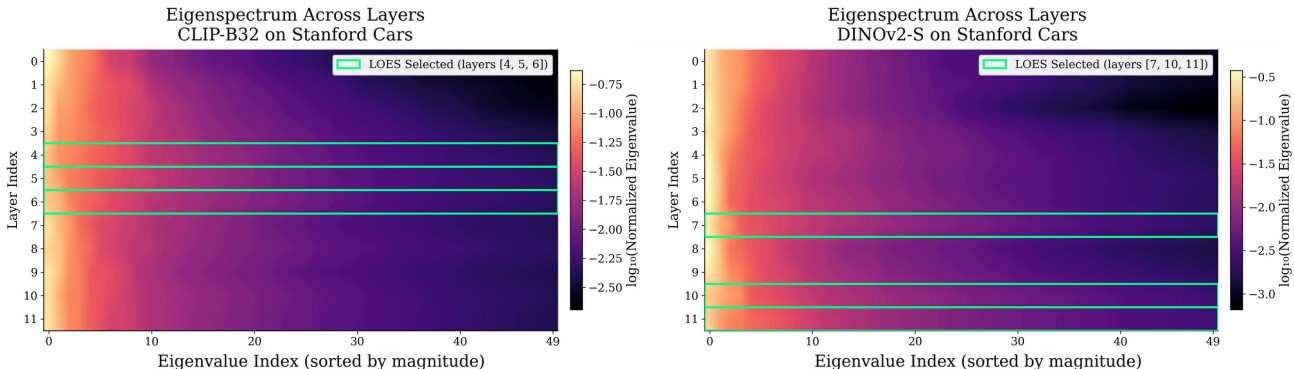

*Figure 10.* Normalized eigenspectrum across encoder layers on Stanford Cars. Each row shows the top-50 eigenvalues (log-scale) of the layer-wise covariance matrix; brighter colors indicate higher eigenvalues. Green borders mark LOES-selected layers. CLIP selects mid-depth layers (4–6) with flatter spectra, while DINOv2 selects later layers (7, 10, 11), reflecting their distinct pretraining paradigms.

### A.6.2. EIGENSPECTRUM ANALYSIS ACROSS LAYERS

Figure 10 visualizes the normalized eigenspectrum of layer-wise covariance matrices, revealing how representation geometry evolves with depth and varies by pretraining paradigm.

CLIP, trained on diverse image-text pairs, exhibits relatively uniform spectra in mid-depth layers, consistent with its early-to-mid layer selection in Table 2. DINOv2, trained via self-distillation on curated images, concentrates isotropic structure in later layers. These spectral patterns provide geometric justification for the layer selection differences: LOES identifies layers where the covariance eigenspectrum is flatter (higher isotropy), yielding better-conditioned ridge probes as established in Theorem A.4.

### A.7. Theoretical Analysis: Geometric Regularization in LOES

The LOES algorithm selects layer residuals by maximizing a hybrid score:

$$\mathcal{S}(l) = -\mathcal{L}_{ridge}(\tilde{\mathbf{x}}_l) - \alpha(1 - \text{Iso}(\tilde{\mathbf{x}}_l)). \tag{12}$$

The first term, $\mathcal{L}_{ridge}$, is intuitive: it ensures the selected feature correlates with the task labels on the available source data. However, the second term, which enforces geometric isotropy, requires rigorous justification. Why should we prefer isotropic residuals over anisotropic ones, even if they yield similar empirical errors?

In this section, we provide this justification from first principles. We posit that for general-purpose representation learning, the goal is not merely to minimize prediction error on a specific dataset (which can lead to overfitting spurious correlations), but to maximize the fidelity of the representation, i.e., its ability to allow a linear probe to recover the true underlying causal mechanism of the task. We prove that maximizing isotropy is mathematically equivalent to minimizing the worst-case structural misalignment (Bias) of the probe.

### A.7.1. PROBLEM FORMULATION

Let $\tilde{\mathbf{x}} \in \mathbb{R}^d$ be the candidate residual feature vector (the component of layer $l$ orthogonal to the current subspace $\mathcal{S}$). We model the downstream task target $y$ as a linear response to these features.

**Assumption A.1** (Linear Mechanism). The target $y \in \mathbb{R}$ is generated by an unknown task weight vector $\mathbf{w}^* \in \mathbb{R}^d$:

$$y = (\mathbf{w}^*)^\top \tilde{\mathbf{x}} + \epsilon, \quad \epsilon \sim \mathcal{N}(0, \sigma^2), \tag{13}$$

where $\tilde{\mathbf{x}}$ is centered with covariance $\mathbf{\Sigma} = \mathbb{E}[\tilde{\mathbf{x}}\tilde{\mathbf{x}}^\top]$.

**Assumption A.2** (Spherical Task Prior). LOES constructs a representation intended to transfer to many unknown downstream tasks. We lack a priori knowledge of which feature directions will be important for future tasks. Therefore, we invoke the *Principle of Maximum Entropy* and assume the task weights $\mathbf{w}^*$ are drawn from a rotationally invariant distribution:

$$\mathbb{E}[\mathbf{w}^*] = \mathbf{0}, \quad \mathbb{E}[\mathbf{w}^*(\mathbf{w}^*)^\top] = \frac{R^2}{d}\mathbf{I}_d. \tag{14}$$

This implies that no direction in the feature space is privileged; the "true" task vector is equally likely to align with any eigenvector of the feature covariance.

A.7.2. SPECTRAL DECOMPOSITION OF FIDELITY

We define the **Fidelity** of the representation as the expected squared Euclidean distance between the estimated weights $\hat{\mathbf{w}}_\lambda$ (learned by a Ridge probe) and the true causal weights $\mathbf{w}^*$. This metric, $\mathcal{E}_{param}$, quantifies the structural correctness of the learned solution.

**Lemma A.3** (Spectral Decomposition of Parameter Error). *The expected squared parameter error decomposes spectrally into two distinct terms: **Alignment Bias** (structural error) and **Estimation Variance** (stochastic error), both dependent on the eigenvalues $\{\mu_i\}_{i=1}^d$ of $\mathbf{\Sigma}$:*

$$\mathcal{E}_{param}(\mathbf{\Sigma}) := \mathbb{E}[\|\hat{\mathbf{w}}_\lambda - \mathbf{w}^*\|^2] = \sum_{i=1}^d \left( \underbrace{\frac{R^2}{d} \frac{\lambda^2}{(\mu_i + \lambda)^2}}_{\text{Alignment Bias } \mathcal{B}(\mu_i)} + \underbrace{\sigma^2 \frac{\mu_i}{(\mu_i + \lambda)^2}}_{\text{Estimation Variance } \mathcal{V}(\mu_i)} \right) \tag{15}$$

*Proof.* **Step 1: Explicit Form of the Estimator Error.** The Ridge Regression estimator is given by $\hat{\mathbf{w}}_\lambda = (\mathbf{\Sigma} + \lambda\mathbf{I})^{-1}\mathbf{\Sigma}_{xy}$. We substitute the generative model for the cross-covariance $\mathbf{\Sigma}_{xy} = \mathbf{\Sigma}\mathbf{w}^* + \mathbf{\Sigma}^{1/2}\mathbf{z}$, where $\mathbf{z}$ is standard Gaussian noise. The error vector $\mathbf{e} = \hat{\mathbf{w}}_\lambda - \mathbf{w}^*$ is:

$$\mathbf{e} = (\mathbf{\Sigma} + \lambda\mathbf{I})^{-1}(\mathbf{\Sigma}\mathbf{w}^* + \mathbf{\Sigma}^{1/2}\mathbf{z}) - \mathbf{w}^* \tag{16}$$

$$= \left[(\mathbf{\Sigma} + \lambda\mathbf{I})^{-1}\mathbf{\Sigma} - \mathbf{I}\right]\mathbf{w}^* + (\mathbf{\Sigma} + \lambda\mathbf{I})^{-1}\mathbf{\Sigma}^{1/2}\mathbf{z}. \tag{17}$$

We simplify the bracketed bias term using the matrix identity $\mathbf{A}^{-1}\mathbf{B} - \mathbf{I} = \mathbf{A}^{-1}(\mathbf{B} - \mathbf{A})$. Setting $\mathbf{A} = \mathbf{\Sigma} + \lambda\mathbf{I}$ and $\mathbf{B} = \mathbf{\Sigma}$, we get $\mathbf{B} - \mathbf{A} = -\lambda\mathbf{I}$. Thus:

$$\mathbf{e} = \underbrace{-\lambda(\mathbf{\Sigma} + \lambda\mathbf{I})^{-1}\mathbf{w}^*}_{\mathbf{e}_{bias}} + \underbrace{(\mathbf{\Sigma} + \lambda\mathbf{I})^{-1}\mathbf{\Sigma}^{1/2}\mathbf{z}}_{\mathbf{e}_{var}}. \tag{18}$$

**Step 2: Computing the Expected Squared Norm.** We seek $\mathbb{E}[\|\mathbf{e}\|^2]$. Since the noise $\mathbf{z}$ has zero mean and is independent of $\mathbf{w}^*$, the cross-term $\mathbb{E}[\mathbf{e}_{bias}^\top\mathbf{e}_{var}]$ vanishes. We analyze the squared norms of the bias and variance terms separately.
*Part A: The Bias Term.*

$$\|\mathbf{e}_{bias}\|^2 = \lambda^2(\mathbf{w}^*)^\top(\mathbf{\Sigma} + \lambda\mathbf{I})^{-2}\mathbf{w}^*. \tag{19}$$

Taking the expectation over the task prior $\mathbf{w}^*$ using the trace trick $\mathbb{E}[\mathbf{u}^\top\mathbf{M}\mathbf{u}] = \text{Tr}(\mathbf{M}\mathbb{E}[\mathbf{u}\mathbf{u}^\top])$:

$$\mathbb{E}_{\mathbf{w}^*}[\|\mathbf{e}_{bias}\|^2] = \lambda^2\text{Tr}\left((\mathbf{\Sigma} + \lambda\mathbf{I})^{-2}\mathbb{E}[\mathbf{w}^*(\mathbf{w}^*)^\top]\right) \tag{20}$$

$$= \lambda^2\text{Tr}\left((\mathbf{\Sigma} + \lambda\mathbf{I})^{-2}\frac{R^2}{d}\mathbf{I}\right) \quad \text{(by Assum. A.2)} \tag{21}$$

$$= \frac{R^2\lambda^2}{d}\sum_{i=1}^d \frac{1}{(\mu_i + \lambda)^2}. \tag{22}$$

*Part B: The Variance Term.*

$$\|\mathbf{e}_{var}\|^2 = \mathbf{z}^\top\mathbf{\Sigma}^{1/2}(\mathbf{\Sigma} + \lambda\mathbf{I})^{-2}\mathbf{\Sigma}^{1/2}\mathbf{z}. \tag{23}$$

Taking the expectation over noise $\mathbf{z}$ with $\mathbb{E}[\mathbf{z}\mathbf{z}^\top] = \sigma^2\mathbf{I}$:

$$\mathbb{E}_{\mathbf{z}}[\|\mathbf{e}_{var}\|^2] = \text{Tr}\left(\mathbf{\Sigma}^{1/2}(\mathbf{\Sigma} + \lambda\mathbf{I})^{-2}\mathbf{\Sigma}^{1/2} \cdot \sigma^2\mathbf{I}\right) \tag{24}$$

$$= \sigma^2\text{Tr}\left(\mathbf{\Sigma}(\mathbf{\Sigma} + \lambda\mathbf{I})^{-2}\right) \quad \text{(cyclic property)} \tag{25}$$

$$= \sigma^2\sum_{i=1}^d \frac{\mu_i}{(\mu_i + \lambda)^2}. \tag{26}$$

Summing Part A and Part B yields Eq. 15. $\qquad\square$

### A.7.3. OPTIMALITY OF ISOTROPY IN LOES

In the context of LOES, we are particularly concerned with the Alignment Bias ($\mathcal{B}$). The variance term $\mathcal{V}$ depends heavily on label noise, which varies by dataset. The bias term, however, represents a fundamental geometric mismatch between our chosen features and the task mechanism. To ensure our representation is robust to *any* task orientation, we must minimize this bias.

We now prove that minimizing the Alignment Bias strictly requires the feature covariance to be isotropic.

**Theorem A.4** (Isotropy Minimizes Alignment Bias). *Let $\mathcal{S}_E = \{\Sigma \succeq 0 \mid Tr(\Sigma) = E\}$ be the set of valid covariance matrices with fixed total signal energy E. The Alignment Bias component of the parameter error is strictly minimized if and only if $\Sigma$ is isotropic.*

$$\arg \min_{\Sigma \in \mathcal{S}_E} \sum_{i=1}^{d} \mathcal{B}(\mu_i) = \frac{E}{d} \mathbf{I}_d \tag{27}$$

*Proof.* **Step 1: Convexity Analysis of the Bias Kernel.** Let $g(\mu) = \frac{1}{(\mu+\lambda)^2}$ be the bias contribution of a single eigenvalue $\mu$ (omitting the positive constant factor $R^2\lambda^2/d$). We compute the first and second derivatives with respect to $\mu$:

$$g'(\mu) = \frac{d}{d\mu}(\mu + \lambda)^{-2} = -2(\mu + \lambda)^{-3} \tag{28}$$

$$g''(\mu) = \frac{d}{d\mu}[-2(\mu + \lambda)^{-3}] = 6(\mu + \lambda)^{-4}. \tag{29}$$

Since the regularization parameter $\lambda > 0$ and eigenvalues $\mu \geq 0$, the term $(\mu + \lambda)^4$ is strictly positive. Consequently, $g''(\mu) > 0$ for all valid $\mu$. This implies that $g(\mu)$ is a **strictly convex function**.

**Step 2: Constrained Optimization via Jensen's Inequality.** We seek to minimize the total bias $J(\boldsymbol{\mu}) = \sum_{i=1}^{d} g(\mu_i)$ subject to the trace constraint $\sum_{i=1}^{d} \mu_i = E$. By **Jensen's Inequality** for convex functions, the average of the function values is bounded below by the function of the average:

$$\frac{1}{d} \sum_{i=1}^{d} g(\mu_i) \geq g\left(\frac{1}{d} \sum_{i=1}^{d} \mu_i\right). \tag{30}$$

Substituting the constraint $\sum \mu_i = E$:

$$\sum_{i=1}^{d} \frac{1}{(\mu_i + \lambda)^2} \geq d \cdot \frac{1}{(E/d + \lambda)^2}. \tag{31}$$

**Step 3: Condition for Strict Optimality.** For a *strictly* convex function like $g(\mu)$, equality in Jensen's Inequality holds if and only if all inputs are identical:

$$\mu_1 = \mu_2 = \cdots = \mu_d = \frac{E}{d}. \tag{32}$$

This spectral condition corresponds uniquely to the isotropic covariance matrix $\Sigma = \frac{E}{d}\mathbf{I}$.

**Conclusion:** Any deviation from isotropy (i.e., spectral skewness) strictly increases the lower bound of the Alignment Bias. Therefore, an isotropic distribution of residual variance is the theoretical optimum for minimizing structural error in the probe. $\square$

### A.7.4. THEORETICAL JUSTIFICATION OF THE LOES OBJECTIVE

This theoretical analysis directly informs the design of the LOES selection criterion $\mathcal{S}(l)$. While the $\mathcal{L}_{ridge}$ term reduces the empirical error on the source task, it cannot guarantee that the features capture the correct causal structure—it is prone to latching onto high-variance spurious correlations. Theorem A.4 establishes that to safeguard against such structural misalignment, the representation must be isotropic. The term $\alpha(1 - \text{Iso}(\tilde{\mathbf{x}}_l))$ in our objective serves as a geometric regularizer that enforces the optimality condition derived in Theorem A.4. By penalizing anisotropy, LOES forces the selected residual features to adopt the optimal geometry for mechanism recovery, ensuring that every selected dimension contributes equally to the representation's transfer capability.

## A.8. Detailed Justification of LOES Objective Components

This subsection provides a detailed justification of the individual components used in the LOES objective. The discussion builds directly on the theoretical results established in the preceding section, with particular emphasis on the role of spectral properties of representations in linear probing. Throughout, isotropy plays a central role as it directly governs the conditioning, stability, and predictability of ridge-based linear evaluation.

### A.8.1. SETTING AND NOTATION

Let $\mathbf{X}_\ell \in \mathbb{R}^{N \times d_\ell}$ denote the centered embedding matrix extracted from layer $\ell$ on a calibration set of $N$ samples, and let $\mathbf{Y} \in \mathbb{R}^{N \times C}$ denote one-hot encoded targets. The objective is to select a subset of layers $S \subset \{1, \ldots, L\}$ with $|S| = K$ such that the resulting representation subspace supports stable and complementary linear prediction.

At each iteration, candidate layers are evaluated using

$$\text{Score}(\ell) = \mathcal{L}_{\text{res}}(\ell) + \alpha\big(1 - \text{Iso}(\mathbf{X}_\ell)\big) + \gamma \text{Red}(\ell) - \eta \text{Tri}(\ell), \tag{33}$$

where each term corresponds to a distinct failure mode identified in the theoretical analysis.

### A.8.2. RESIDUAL RIDGE LOSS AND SPECTRAL SENSITIVITY

For a feature matrix $\mathbf{X}$ and targets $\mathbf{Y}$, LOES relies on ridge regression,

$$\hat{\mathbf{W}} = \arg\min_{\mathbf{W}} \|\mathbf{X}\mathbf{W} - \mathbf{Y}\|_F^2 + \lambda\|\mathbf{W}\|_F^2, \qquad \hat{\mathbf{W}} = (\mathbf{X}^\top\mathbf{X} + \lambda\mathbf{I})^{-1}\mathbf{X}^\top\mathbf{Y}. \tag{34}$$

As shown in Theorem A.4, the expected error of the ridge estimator depends not only on the alignment between $\mathbf{Y}$ and the feature subspace, but also on the distribution of eigenvalues of $\mathbf{X}^\top\mathbf{X}$. In particular, highly anisotropic spectra lead to estimators whose behavior is dominated by a small number of principal directions, while directions associated with smaller eigenvalues contribute disproportionately to variance.

Let $\widehat{\mathbf{Y}}$ denote the cumulative prediction from already selected layers, and define the residual

$$\mathbf{R} = \mathbf{Y} - \widehat{\mathbf{Y}}.$$

The residual ridge loss $\mathcal{L}_{\text{res}}(\ell)$ evaluates how effectively a candidate layer explains this residual signal. Importantly, residuals often emphasize directions with lower variance in the feature space, making their estimation particularly sensitive to anisotropy. This sensitivity motivates the inclusion of an explicit isotropy term alongside residual fitting.

### A.8.3. ORTHOGONALIZATION, REDUNDANCY, AND SPECTRAL OVERLAP

Let $\mathbf{X}_S$ denote the concatenation of features from the selected layers. LOES evaluates each candidate layer after removing its projection onto the current subspace:

$$\widetilde{\mathbf{X}}_\ell = \mathbf{X}_\ell - \mathbf{X}_S(\mathbf{X}_S^\top\mathbf{X}_S + \varepsilon\mathbf{I})^{-1}\mathbf{X}_S^\top\mathbf{X}_\ell. \tag{35}$$

This operation isolates directions that are not already represented. From a spectral perspective, it prevents repeated selection of layers whose dominant eigen-directions coincide with those already chosen. This is particularly important when individual layers are anisotropic, as their strongest directions may otherwise dominate the selection process despite offering little new information.

Residual redundancy is further quantified by

$$\text{Red}(\ell) = \max_{j \in S} \frac{\|\mathbf{X}_\ell^\top\mathbf{X}_j\|_F}{\|\mathbf{X}_\ell\|_F \|\mathbf{X}_j\|_F}, \tag{36}$$

which explicitly penalizes candidates whose feature directions remain strongly aligned with those already selected, even after orthogonalization.

### A.8.4. ISOTROPY AND CONDITIONING OF LINEAR PROBES

Let $\mathbf{\Sigma}_\ell = \frac{1}{N}\mathbf{X}_\ell^\top\mathbf{X}_\ell$ with eigenvalues $\{\mu_i\}$. LOES measures isotropy via

$$\text{Iso}(\mathbf{X}_\ell) = \frac{\bar{\mu}}{\sqrt{\text{Var}(\{\mu_i\}) + \delta}}, \qquad \bar{\mu} = \frac{1}{d_\ell}\sum_i \mu_i. \tag{37}$$

This quantity decreases as the covariance spectrum becomes more uneven. As established in Theorem A.4, representations with larger spectral variance admit ridge estimators whose risk is more sensitive to the orientation of the target relative to

dominant eigen-directions. Conversely, representations with more uniform spectra exhibit more consistent behavior across tasks and sample realizations.

In the context of LOES, isotropy does not serve as a proxy for task relevance. Instead, it acts as a regularizer on the selection process by favoring layers whose linear probes are better conditioned. This is particularly important in the greedy setting, where early selection of highly anisotropic layers can distort subsequent residual estimates and lead to unstable layer rankings.

### A.8.5. CLASS GEOMETRY

For classification tasks, LOES additionally evaluates the geometry of class centroids computed from $\widetilde{\mathbf{X}}_\ell$. Let $\boldsymbol{\mu}_c$ denote the centroid of class $c$. The class geometry term is

$$\text{Tri}(\ell) = \mathbb{E}_{(a,b,c)}\left[\tfrac{1}{2}\sqrt{\|\boldsymbol{\mu}_b - \boldsymbol{\mu}_a\|^2\|\boldsymbol{\mu}_c - \boldsymbol{\mu}_a\|^2 - \langle\boldsymbol{\mu}_b - \boldsymbol{\mu}_a, \boldsymbol{\mu}_c - \boldsymbol{\mu}_a\rangle^2}\right]. \tag{38}$$

This term penalizes representations in which class means collapse into low-dimensional affine configurations. Such collapse is often correlated with strong anisotropy, where only a small number of directions dominate both variance and class separation.

Overall, isotropy interacts with all other components of the LOES objective. It stabilizes ridge estimation, moderates the effects of residual fitting, reduces the influence of dominant but redundant directions, and supports more balanced class geometry. For these reasons, isotropy serves as a structural regularizer within LOES rather than a standalone selection criterion.

### A.9. Time Complexity Analysis of LOES

We analyze the computational complexity of LOES, focusing on the calibration and layer-selection stages, which constitute the only additional overhead introduced beyond standard frozen-encoder transfer learning.

**Notation.** Let $L$ denote the total number of encoder layers, $d$ the embedding dimension per layer, $C$ the number of classes, $N_{\text{cal}}$ the number of calibration samples, and $K \ll L$ the number of layers selected by LOES. All encoder parameters remain frozen during calibration.

**Calibration Feature Extraction.** LOES first extracts intermediate representations from all encoder layers for $N_{\text{cal}}$ samples. This requires a single forward pass through the encoder per batch and incurs

$$\mathcal{O}(N_{\text{cal}} \cdot L \cdot d)$$

time and

$$\mathcal{O}(L \cdot N_{\text{cal}} \cdot d)$$

memory to store pooled layer representations. This cost is incurred once per dataset.

**Per-Layer Scoring.** Each layer is scored independently using a closed-form linear probe and a geometric regularization term. Solving the linear system dominates this step and requires

$$\mathcal{O}(d^3 + N_{\text{cal}} \cdot d^2)$$

per layer. Aggregated across all layers, the total cost is

$$\mathcal{O}\big(L \cdot (d^3 + N_{\text{cal}} \cdot d^2)\big).$$

**Greedy Complementary Selection.** LOES selects $K$ layers using a greedy procedure that evaluates candidate layers against the current selection. At each iteration, all remaining layers are considered, and scoring involves residual fitting, orthogonalization with respect to previously selected layers, and redundancy-aware geometric penalties. Each iteration incurs

$$\mathcal{O}\big(L \cdot (d^3 + N_{\text{cal}} \cdot d^2)\big),$$

leading to a total selection cost of

$$\mathcal{O}\big(K \cdot L \cdot (d^3 + N_{\text{cal}} \cdot d^2)\big).$$

**Overall Complexity.** Combining all stages, the total time complexity of LOES is

$$\mathcal{O}\big(N_{\text{cal}} \cdot L \cdot d \ + \ K \cdot L \cdot (d^3 + N_{\text{cal}} \cdot d^2)\big),$$

while the memory complexity is dominated by stored calibration features and scales as

$$\mathcal{O}(L \cdot N_{\text{cal}} \cdot d).$$

**Practical Implications.** In practice, LOES is computationally efficient because calibration sets are small, the number of selected layers is limited ($K \leq 4$ in all experiments), and all operations are performed on frozen representations. Consequently, the overhead of LOES is negligible compared to encoder pretraining or downstream fine-tuning and typically completes within seconds on a single GPU for standard vision and language encoders.

### A.10. GeoReg

In addition to the layer selection procedure, some experiments incorporate a geometric regularization term during probe training to mildly constrain the geometry of the fused representation. This term, referred to as GeoReg in the implementation, is controlled by a single scalar hyperparameter that multiplies the entire regularization contribution. In all reported experiments, this weight is fixed to a small value of $0.1$ when enabled, and set to zero otherwise. The design choice reflects the intended role of the regularizer as a secondary bias rather than a competing objective. Concretely, GeoReg combines two effects computed on the current batch features: a spectral dispersion term given by the variance of the eigenvalues of the feature covariance matrix, and a class-level separation term based on the logarithm of the area of a simplex formed by randomly sampled class centroids. The spectral component penalizes highly anisotropic representations by increasing loss when variance concentrates along a few directions, while the centroid-based term encourages non-degenerate class configurations by favoring larger geometric separation.

The relative scaling of these components is fixed in the code, leaving the outer weight as the only tunable hyperparameter. Empirically, larger values were found to interfere with optimization of the primary cross-entropy objective, especially in low-data or high-class-count regimes, while smaller values had negligible effect. The chosen value therefore reflects a compromise that consistently biases training toward better-conditioned feature spaces without destabilizing learning. Importantly, the regularizer is applied only when the number of classes is sufficient to define meaningful centroid geometry, and it is automatically disabled otherwise. This conditional use ensures that GeoReg does not introduce spurious gradients in degenerate or low-class settings. Overall, the hyperparameterization mirrors the broader philosophy of the method: geometric criteria are used to gently shape representation structure, while predictive performance remains primarily driven by the ridge-regularized linear probe.

### A.11. GeoReg Computation Overhead

GeoReg adds $\sim$5.6% per-batch overhead on DINOv2-S/14, Stanford Cars (trainable, $K = 3$), dominated by eigenvalue decomposition of the $D$-dimensional covariance matrix (11ms for $D = 768$; see Table A15).

| Component | Without GeoReg | With GeoReg |
|---|---|---|
| Forward | 102.9 ms | 103.4 ms |
| Loss | 0.3 ms | 11.4 ms |
| Backward | 230.9 ms | 238.6 ms |
| Step | 8.5 ms | 8.5 ms |
| Total | 342.7 ms | 361.8 ms (+5.6%) |

*Table A15.* Per-batch computational overhead of GeoReg on DINOv2-S/14 (Stanford Cars, trainable, $k = 3$). The overhead is modest (+5.6%) and arises primarily from covariance eigendecomposition.

### A.12. Additional Implementation Details of Geometric Regularization

This subsection provides additional details on the geometric regularization term used during downstream adaptation, focusing on design and implementation choices that are not covered in the main text. The goal of this regularizer is to gently bias learned representations toward favorable linear geometry, consistent with the ridge-based analysis and layer selection criteria, without introducing task-specific constraints or significant optimization overhead.

The regularizer is applied to the fused representation produced after layer aggregation and optional adapter projection. Let $Z \in \mathbb{R}^{B \times d}$ denote the batch of fused features, where $B$ is the batch size. The geometric penalty is added to the task loss

with a fixed scalar weight and is evaluated independently for each mini-batch. Importantly, the regularizer operates on batch-level statistics and does not require storing running estimates or dataset-level covariance statistics.

The isotropy component is computed using the empirical batch covariance

$$\Sigma_Z = \frac{1}{B}(Z - \bar{Z})^\top (Z - \bar{Z}),$$

where $\bar{Z}$ denotes the batch mean. Rather than normalizing the spectrum by trace or enforcing a hard constraint on eigenvalues, the implementation penalizes the variance of the eigenvalue spectrum. This choice directly targets spectral concentration while remaining numerically stable and differentiable. In practice, a small diagonal jitter is added to $\Sigma_Z$ to avoid numerical issues when batch size is small or features are nearly collinear. This formulation aligns with the theoretical analysis linking isotropy to reduced worst-case alignment bias in ridge regression, while avoiding additional normalization hyperparameters.

The class-geometry component is implemented using batch-level class centroids. When at least three distinct classes are present in a batch, three centroids are sampled uniformly at random and the area of the triangle they form is computed. This computation is intentionally stochastic and lightweight: only a single triplet is used per batch, and no attempt is made to enumerate or average over all possible triplets. The logarithm of the resulting area is incorporated into the loss with a negative sign, discouraging degenerate or nearly collinear class configurations. If fewer than three classes are present in a batch, this term is skipped entirely. This conditional behavior ensures that the regularizer does not introduce spurious gradients in low-class or imbalanced settings.

Both components are combined under a single weighting coefficient. Across all experiments, this coefficient is fixed to a small value and is not tuned per dataset or model. Empirically, larger values were observed to interfere with optimization of the primary task objective, while smaller values had negligible effect. The chosen value therefore reflects a conservative trade-off, where geometric structure is encouraged without dominating the training dynamics. Notably, even when this hyperparameter is suboptimal, improvements over the last-layer baseline are consistently observed, indicating that the method is not overly sensitive to precise tuning.

When the backbone encoder is frozen, gradients from the geometric regularizer affect only the probe and any adapter layers. When the encoder is trainable, the regularizer additionally stabilizes representation geometry during fine-tuning, mitigating collapse of variance into a small number of directions. For dense prediction and regression tasks, where class centroids are ill-defined or computationally expensive to estimate, the centroid-based term is disabled and only the isotropy component is retained.

---

**Algorithm 1** LOES Layer Selection Algorithm (Classification)

---

**Input:** Embeddings $\mathcal{E} = \{X_1, \ldots, X_L\}$, labels $y$, budget $K$, parameters $\lambda, \alpha, \gamma, \eta$
**Output:** Selected layer indices $\mathcal{S}$
$Y \leftarrow \mathrm{OneHot}(y)$
$\mathcal{S} \leftarrow \emptyset, \quad X_\mathcal{S} \leftarrow \emptyset, \quad \widehat{Y} \leftarrow \mathbf{0}$
{— Stage (i): Initial Selection (raw features, original targets) —}
$s^\star \leftarrow \arg\min_l \left[ \mathcal{L}_{\mathrm{ridge}}(X_l, Y) + \alpha\big(1 - \mathrm{Iso}(X_l)\big) \right]$ {Eq. (1), (4)}
$\mathcal{S} \leftarrow \{s^\star\}, X_\mathcal{S} \leftarrow X_{s^\star}$
$\widehat{Y} \leftarrow \mathrm{RidgePred}(X_{s^\star}, Y)$ {refit on raw $X_{s^\star}$ vs $Y$}
$R \leftarrow Y - \widehat{Y}$ {residual; see Eq. (7) discussion}
{— Stage (ii): Greedy Complementary Selection —}
**while** $|\mathcal{S}| < K$ **do**
   $v_{\mathrm{best}} \leftarrow \infty, \quad s^\star \leftarrow$ None
   **for** $l \in \{1, \ldots, L\} \setminus \mathcal{S}$ **do**
      {Orthogonalize $X_l$ w.r.t. selected context $X_\mathcal{S}$ – Eq. (3)}
      $X_l^c \leftarrow X_l - \mu(X_l), \quad X_\mathcal{S}^c \leftarrow X_\mathcal{S} - \mu(X_\mathcal{S})$
      $\widetilde{X}_l \leftarrow X_l^c - X_\mathcal{S}^c\big((X_\mathcal{S}^c)^\top X_\mathcal{S}^c + \epsilon I\big)^{-1}(X_\mathcal{S}^c)^\top X_l^c + \mu(X_l)$
      {Score components – Eq. (7)}
      $\mathcal{L} \leftarrow \mathcal{L}_{\mathrm{ridge}}(\widetilde{X}_l, R)$ {Loss$_\ell$: marginal fit on residual}
      $\rho \leftarrow \max_{j \in \mathcal{S}} \dfrac{\|X_l^\top X_j\|_F}{\|X_l\|_F \|X_j\|_F}$ {Red$_\ell$, Eq. (5)}
      $\tau \leftarrow \mathrm{TriGeo}(\widetilde{X}_l, y)$ {Tri, Eq. (6); $\eta=0$ if regression}
      $v \leftarrow \mathcal{L} + \alpha\big(1 - \mathrm{Iso}(X_l)\big) + \gamma\rho - \eta\tau$ {Eq. (7)}
      **if** $v < v_{\mathrm{best}}$ **then**
         $v_{\mathrm{best}} \leftarrow v, \quad s^\star \leftarrow l$
      **end if**
   **end for**
   **if** $s^\star =$ None **then**
      **break**
   **end if**
   {— Refit on raw features against original targets and update ensemble —}
   $\widehat{Y} \leftarrow \widehat{Y} + \mathrm{RidgePred}(X_{s^\star}, Y)$ {refit uses $(X_{s^\star}, Y)$, not $(\widetilde{X}_{s^\star}, R)$}
   $R \leftarrow Y - \widehat{Y}$
   $\mathcal{S} \leftarrow \mathcal{S} \cup \{s^\star\}, \quad X_\mathcal{S} \leftarrow [X_\mathcal{S}, X_{s^\star}]$
**end while**
**return** $\mathcal{S}$

---

| Task | Dataset | Modality | Samples | Split |
|------|---------|----------|---------|-------|
| **List of Datasets Used** | | | | |
| Audio Classification | ASVspoof 2019 (LA) | Audio | 121,461 | Train/Dev/Test |
| | CREMA-D | Audio-Visual | 7,442 | Train/Test |
| | Google Speech Commands | Audio | 105,829 | Train/Val/Test |
| Segmentation | COCO-Stuff 164K | Image | 164,000 | Train/Test |
| | Cityscapes | Image | 5,000 | Train/Val/Test |
| Image Classification | Stanford Cars (196) | Image | 16,185 | Train/Test |
| | Mini-ImageNet | Image | 60,000 | Train/Val/Test |
| | Stanford Dogs | Image | 20,580 | Train/Test |
| | CUB-200-2011 | Image | 11,788 | Train/Test |
| | CIFAR-100 | Image | 60,000 | Train/Test |
| | SUN397 | Image | 108,754 | Train/Test |
| | ImageNet | Image | 1,281,167 | Train/Val/Test |
| Text Classification | MTOP Domain | Text | 15,667 | Train/Val/Test |
| | Twitter Emotion | Text | 20,000 | Train/Val/Test |
| | Amazon Counterfactual | Text | 4,000 | Train/Test |
| | MASSIVE Scenario | Text | 16,521 | Train/Val/Test |
| | MASSIVE Intent | Text | 16,521 | Train/Val/Test |
| | Tweet Sentiment Extraction | Text | 31,015 | Train/Test |
| Multimodal | Amazon Products-2023 | Image/Text | 140,000 | Train/Val/Test |
| | Fashion-Gen | Image/Text | 67,306 | Train/Val |
| | Fakeddit | Multimodal | 140,000 | Train/Val/Test |

*Table A16.* **Comprehensive overview of benchmark datasets across modalities.** This table details the tasks, modalities, and sample distributions for the datasets utilized in this study, including benchmarks from the MTEB suite and fine-grained vision tasks.

## Datasets

Below we provide detailed descriptions of the datasets (Appendix Table A16) utilized in our experiments, categorized by their primary modality.

### Audio Classification Datasets

- **ASVspoof 2019 Logical Access (LA):** A standard benchmark derived from the VCTK corpus for detecting logical access attacks (Text-to-Speech and Voice Conversion) against speaker verification systems. It includes genuine speech alongside spoofing attacks generated by 19 different algorithms.

- **CREMA-D:** The Crowd-sourced Emotional Multimodal Actors Dataset features facial and vocal expressions from 91 actors of varying ages and ethnicities. It contains audio-visual clips of 12 fixed sentences spoken with six different emotions (Anger, Disgust, Fear, Happy, Neutral, Sad).

- **Google Speech Commands (V2):** A large-scale benchmark for Keyword Spotting (KWS) consisting of one-second audio clips of 35 specific spoken words. It is designed to train small-footprint models for on-device applications.

### Segmentation Datasets

- **COCO-Stuff 164K:** A large-scale scene understanding dataset that extends COCO 2017. Unlike the standard COCO which focuses on "things" (countable objects), COCO-Stuff provides dense pixel-wise annotations for 172 classes, including 91 "stuff" classes (amorphous regions like sky, grass, and water).

- **Cityscapes:** A premier dataset for semantic urban scene understanding. It features high-quality pixel-level annotations of complex street scenes from 50 different cities, capturing diverse traffic situations and scene layouts.

### Image Classification Datasets

- **Stanford Cars (Cars196):** A fine-grained classification benchmark containing 196 classes of vehicles defined by Make, Model, and Year (e.g., distinguishing a "2011 BMW M3" from a "2012 BMW M3").

- **Mini-ImageNet:** A subset of ILSVRC-12 designed for few-shot learning and rapid prototyping. It consists of 100 classes selected from the larger ImageNet dataset, typically resized to lower resolutions for meta-learning tasks.

- **Stanford Dogs:** A fine-grained dataset subset from ImageNet, challenging models to distinguish between 120 closely related dog breeds.

- **CUB-200-2011:** The Caltech-UCSD Birds dataset is a benchmark for fine-grained visual categorization. It contains 200 bird species and includes rich annotations such as bounding boxes, part locations, and binary attributes.

- **CIFAR-100:** A widely used benchmark containing low-resolution ($32 \times 32$) images across 100 classes. The classes are hierarchically organized into 20 superclasses, making it suitable for testing hierarchical learning.

- **SUN397:** A definitive benchmark for scene recognition rather than object classification. It covers 397 distinct scene categories (e.g., "Abbey," "Diner," "Forest") from the larger SUN Database.

- **ImageNet-1k:** A large-scale dataset for object recognition containing 1,000 object categories. It is a subset of the larger ImageNet database and follows the WordNet hierarchy, containing a variety of animal, plant, and object classes.

**Text Classification Datasets (MTEB)**

- **MTOP Domain:** A task-oriented semantic parsing dataset used to classify user utterances into 11 specific action domains (e.g., Alarm, Messaging, Weather).

- **Twitter Emotion:** Also known as the MTEB Emotion dataset, this consists of English tweets labeled with six basic emotions to evaluate how well embeddings capture emotional nuance.

- **Amazon Counterfactual:** A dataset designed to challenge models in identifying counterfactual statements within product reviews, distinguishing between actual events and hypothetical scenarios.

- **Amazon MASSIVE Scenario:** A Natural Language Understanding task where utterances are classified into 18 broad functional categories.

- **Amazon MASSIVE Intent:** A more granular NLU task requiring the classification of user utterances into 60 specific action-based goals.

- **Tweet Sentiment Extraction:** A Natural Language Processing task where, given a social media post and its overall sentiment (positive, negative, or neutral), the objective is to extract the specific word or phrase from the text that best reflects and supports that sentiment

**Multimodal Datasets**

- **Amazon Products-2023 (Amazon-M2):** A stratified sampling strategy was applied to obtain 140K products, covering all 248 categories with balanced category sizes when possible and full inclusion of low-frequency categories.

- **Fashion-Gen:** A dataset for high-resolution text-to-image synthesis in the fashion domain, pairing professional images with stylist-written descriptions. We filtered for unique product IDs, resulting in 67,306 samples.

- **Fakeddit:** A massive multimodal fake news detection benchmark from Reddit. It pairs text posts with images and metadata to detect various types of misinformation. We randomly sampled 140K image-text pairs for our experiments.

