# OpenReview forum: "Uncovering the Latent Potential of Deep Intermediate Representations"
_ICML.cc/2026/Conference — ICML 2026 spotlight_

### Official Review · Reviewer_DEiU · 2026-03-10

**Soundness:** 3
**Presentation:** 3
**Significance:** 3
**Originality:** 3
**Overall Recommendation:** 5
**Confidence:** 3

**Summary:**

This paper introduces LOES, a greedy spectral method that selects a small subset of intermediate layers from pretrained encoders by minimizing residual error under orthogonality and isotropy constraints. A companion fine-tuning loss, GeoReg (Eq. 9), regularizes  spectral imbalance in the embedding covariance and encourages class separation of the fused representation. Experiments span vision, language (non autoregressive), and speech encoders on classification, regression, segmentation, and cross-lingual tasks, consistently outperforming last-layer and learnable-weight baselines.

**Compliance With Llm Reviewing Policy:**

Affirmed.

**Final Justification:**

I thank the authors for the thorough rebuttal. The exhaustive search comparison, the revised k=3-4 framing, and the stronger learned-concat baseline address my main concerns. Therefore, I maintain my positive score. I expect all promised baselines and table corrections to be fully integrated into the camera-ready version.

**Key Questions For Authors:**

1. The hyperparameter ablation (Table A1) covers only MTOP/ModernBERT. Are alpha=1.0, gamma=0.5, eta=0.1 similarly stable on a vision task or a speech task?

2. For k=3 from L=12 layers, exhaustive search over C(12,3)=220 subsets is feasible. How often does the greedy LOES solution match the globally optimal subset, and how large is the gap when it does not?

3. Table A4 reports t=-6.57 for CLIP-B32 / S. Dogs (implying LOES underperforms), but Table 3 shows LOES (61.75) > Last (60.65). Can you clarify this discrepancy? For cases where LOES genuinely underperforms (DeiT-B/16 on Mini-IN and S. Dogs), what model or task properties predict failure?

4. Table A3 shows BERT-base accuracy climbing from 96.08% (k=1) to 98.43% (k=10) with no saturation. Why should the default value for k be in $\{3,4\}$ rather than tuning k per model?

5. The learnable-weights baseline uses a scalar weighted sum, which is less expressive than LOES's concatenation structure. A stronger learned  baseline (e.g., per-layer linear projections to a low dimension followed by concatenation) would isolate whether gains come from the selection criterion itself or from the richer concatenated representation.

**Limitations:**

Yes

**Strengths And Weaknesses:**

## Strengths

S1. **Statistical rigor.** Results are mean +/- std over 5 seeds. Paired statistical significance tests (Table A4) confirm significance for the large majority of model-dataset pairs, and one-way ANOVA (Table A5) confirms that encoder choice remains a significant factor under LOES.

S2. **Breadth of evaluation.** The paper covers an incredibly vast variety of downstream tasks, e.g,  fine-grained visual classification, general classification, text benchmarks, speech tasks, regression, dense prediction, and cross-lingual transfer. The speech gains are convincing: CREMA-D improves from 37.84% to 69.06% and ASVspoof from 90.89% to 98.80% (Table A9). On the language side, ModernBERT on MTOP jumps from 78.07% to 94.48% (+16.4 points absolute, Table 5), one of the largest gains reported.

S3. **Low sensitivity to calibration budget and hyperparameters.** Table A1 shows 64 calibration samples yield 95.90% on MTOP vs. 95.96% at 512. The fixed configuration (alpha=1.0, gamma=0.5, eta=0.1/0.0) works across all experiments without task-specific tuning. Table A2 shows that even 1% calibration data (Avg 64.07) is competitive, with performance approaching the peak (65.62 at 50%) by 30% (Avg 65.00); the paper fixes calibration at 20% for all experiments.

S4. **Interpretable layer selection patterns.** Table 2 and Figure 4 reveal that encoders pretrained on diverse corpora (CLIP, DINOv2) select early-to-middle layers and distribute task-relevant information across depth, while narrowly pretrained models (MAE, ViT-IN21k) concentrate it in final layers. This has direct practical value for choosing adaptation strategies.

S5. **Negligible computational overhead.** Table 1 shows LOES adds minimal wall-clock time: SUN397 with DINOv2-S/14 increases from 2259s to 2307s (+2.1%); MTOP with ModernBERT from 529s to 660s (+24.8%, dominated by the larger fused representation). Parameter increase is under 1%.

## Weaknesses

W1. **Hyperparameter ablation is narrow.** Table A1 covers only MTOP with ModernBERT-base. The authors fix a single configuration across all modalities and claim generality, but this is validated on only one task. Sensitivity analysis on at least one vision and one speech task would strengthen the claim.

W2. **No comparison against the globally optimal subset.** For k=3 from L=12 layers, exhaustive search over C(12,3)=220 subsets is feasible for ablation studies. The authors acknowledge greedy suboptimality (Appendix A.2) but never report how often the greedy solution matches the global optimum.

W3. **Inconsistent gains and an internal data conflict.** Table A4 shows LOES significantly underperforms on DeiT-B/16 / Mini-IN (t=-7.96) and DeiT-B/16 / S. Dogs (t=-9.38), confirmed by Table 3. However, Table A4 also reports t=-6.57 for CLIP-B32 / S. Dogs (implying LOES underperforms), yet Table 3 shows LOES (61.75) beating Last (60.65), an apparent sign error or data mismatch. The paper does not characterize when LOES is expected to fail.

W4. **Table A3 does not uniformly support the k=3-4 recommendation.** The paper states performance "typically saturates after k=3-4" (Section 5). For ModernBERT-base on MTOP, performance peaks at k=5 (96.01%) before declining. For BERT-base, accuracy rises nearly monotonically from k=1 (96.08%) to k=10 (98.43%) with no clear saturation. The universal recommendation of $k \in \{3,4\}$ is not fully supported by the paper's own data.

---

> ### Author Rebuttal · Authors · 2026-03-30
>
> We thank Reviewer DEiU for the positive assessment, and for recognizing our statistical rigor, breadth of evaluation, and low calibration sensitivity. We address each point below and all the changes will be incorporated in the final revision.
>
> ## W1. Hyperparameter ablation is narrow (+ Q1)
>
> We provide cross-modal validation. On audio (ASVspoof LA19, Wav2Vec 2.0, K=4):
>
> | Method     | α   | γ   | η    | K | ncal | Layers Selected | Test Acc ↑ |
> |------------|-----|-----|------|---|------|------------------|-------------|
> | Last Layer | -   | -   | -    | 1 | -    | -                | 0.90891     |
> | Last-4 Concat | - | - | - | 4| - | - | 94.83%
> | LOES       | 0   | 0   | 0    | 4 | 512  | 8, 11, 12, 10    | 0.95526     |
> | LOES       | 0   | 0.1 | 0    | 4 | 512  | 8, 0, 1, 2       | 0.97904     |
> | LOES       | 0   | 0.5 | 0    | 4 | 512  | 8, 0, 1, 2       | 0.97904     |
> | LOES       | 1   | 0.5 | 0.1  | 4 | 64   | 1, 0, 3, 2       | 0.98133     |
> | LOES       | 1   | 0.5 | 0.1  | 4 | 512  | 1, 0, 3, 2       | 0.98301     |
>
> LOES outperforms all baselines with <0.3pp variation across configurations. For vision, the fixed default improves over last-layer on all 7 benchmarks (Table 3). We additionally provide a 2D $\alpha \times \gamma$ heatmap on MTOP (MBERT-B, K=4): https://tinyurl.com/3acx74ry
>
> The surface forms a broad plateau: default ($\alpha$=1.0, $\gamma$=0.5) achieves 95.90%, within 0.25pp of the optimum (96.15%). Even the worst grid configuration (94.80%) outperforms last-layer (81.37%) by 13+ points.
>
> ## W2. No comparison against globally optimal subset (+ Q2)
>
> We exhaustively trained all $C(12,3)=220$ subsets for BERT-B (frozen), evaluating each on the test set:
>
> **MTEB MTOP:**
>
> | | Layers | Test Acc |
> |---|---|---|
> | Global Optimal | [3, 6, 11] | **97.83%** |
> | LOES | [11, 10, 6] | 96.67% |
> | Last Layer | [11] | 95.51% |
>
> **MTEB Emotion:**
>
> | | Layers | Test Acc |
> |---|---|---|
> | Global Optimal | [3, 10, 11] | **71.08%** |
> | LOES | [11, 10, 9] | 69.39% |
> | Last Layer | [11] | 68.40% |
>
> LOES achieves within 1.16pp (MTOP) and 1.69pp (Emotion) of the globally optimal subset, with 2/3 layers common between them. Finding this optimum required evaluating all 220 possible 3-layer combinations, taking several hours of GPU compute even for BERT-base (12 layers). In contrast, LOES uses a few closed-form computations on a small calibration set followed by a single training run, avoiding any combinatorial search.
>
> For MBERT (22 layers), exhaustive search is infeasible and would require $C(22,3)=1{,}540$ runs; for BERT-L (24 layers), $C(24,3)=2{,}024$, for these large models LOES provides the largest gains (Table 5). LOES offers a principled and scalable alternative that achieves near-optimal layer selection at a fraction of the cost.
>
> ## W3. Inconsistent gains and internal data conflict (+ Q3)
>
> We thank the reviewer for catching this discrepancy. The t-statistic of -6.57 reported in Table A4 for CLIP-B32 / S. Dogs was incorrect due to a sign error in our reporting script. The correct value is t=+3.60 (p=0.023), confirming that LOES outperforms the last-layer baseline on this combination, consistent with Table 3 (61.75% vs. 60.65%). We will correct Table A4 in the revision.
>
> For cases where LOES genuinely underperforms (DeiT-B/16 on Mini-IN and S. Dogs), the differences are very small in absolute terms (0.19% and 0.25% respectively). DeiT-B/16 is pretrained via supervised distillation exclusively on ImageNet variants, which causes task-discriminative information to concentrate heavily in the final layers. In such cases, the final layer is already near-optimal and intermediate layers contribute noise rather than complementary signal, leaving little room for LOES to improve. We discuss the limitations in Appendix A.2.
>
> ## W4. k=3-4 recommendation (+ Q4)
>
> We acknowledge our phrasing conveyed the wrong meaning, what we intended is that LOES performance starting from k=3-4 is typically better than the last layer or last 3-4 concatenated layers. For ModernBERT: k=3 (95.99%) to k=5 (96.01%) gains 0.02pp at 67% higher dimension. For BERT-base: k=4 (97.74%) to k=10 (98.43%) gains 0.69pp at 2.5x dimension. We will revise to: *"For most settings, $k \in \{3,4\}$ offers the best accuracy-efficiency trade-off."*
>
> ## Q5. Stronger baseline
>
> We thank the reviewer for this suggestion. We implemented the stronger baseline (per-layer linear projections to low dimension followed by concatenation of all L layers) on MBERT-B:
>
> | Method | MTOP | AM-Scenario |
> |---|---|---|
> | Last Layer | 80.96 | 63.45 |
> | Learnable Concat (all 22 layers) | 93.91 | 77.51 |
> | LOES-4 | 94.19 | 78.51 |
>
> LOES-4 outperforms the stronger Learnable Concat baseline on both tasks, despite the latter using all 22 layers with per-layer projections (1.7x more parameters and FLOPs at inference). This indicates that the gains come from the selection criterion itself: LOES matches or exceeds a fully parameterized multi-layer fusion at a fraction of the cost.

---

> > ### Author Rebuttal · Reviewer_DEiU · 2026-04-01
> >
> > I thank the authors for the thorough rebuttal. The exhaustive search comparison, the revised k=3-4 framing, and the stronger learned-concat baseline  address my main  concerns. Therefore, I maintain my positive score.

---

### Official Review · Reviewer_mPEw · 2026-03-10

**Soundness:** 3
**Presentation:** 4
**Significance:** 4
**Originality:** 3
**Overall Recommendation:** 5
**Confidence:** 3

**Summary:**

While it is knows the internal activations can be a good representation for different tasks, efficiency selecting the correct layer is an open challenge. This approach presents a set of heuristics for efficient selection.  Extensive experiments justify the effectiveness of the approach.

**Compliance With Llm Reviewing Policy:**

Affirmed.

**Key Questions For Authors:**

Can you compute the expensive  upper bound and show how far these heuristics are from this bound?
Can you provide a scaling law for search compute vs final prediction accuracy?

**Limitations:**

Yes

**Strengths And Weaknesses:**

Overall, I have a positive opinion of this paper. This is due to several strengths: i) the task of selecting the best combination of activations is highly practical and common in ML. Most practitioners default to a lazy choice like choosing the last layer for lack of better alternatives. Proposing a more efficient choice is threfore significant and timely. ii) the set of heuristics is non-trivial. While the redundancy metric seems obvious, isometry and separability are interesting and their precise formulation non-trivial iii) the finetuning losses provide an extra contribution iv) the experiments are quite extensive with significant ablations for all heuristic on many models and datasets. The gains over lazier options are signifcant and well worth the effort.

The main limitation is the hyper-parameter calibration which seems somewhat sensitive. This is mentioned in the paper.

---

> ### Author Rebuttal · Authors · 2026-03-30
>
> We thank Reviewer mPEw for the positive and insightful review. We address the question and the identified limitation directly. The raised concerns have strengthened the paper and all the above changes will be incorporated in the final revision.
>
> ## Limitation: Hyperparameter calibration seems somewhat sensitive
>
> We acknowledge this limitation (Appendix A.2) while emphasizing practical robustness. To address the interaction concern directly, we provide a 2D sensitivity analysis of α (isotropy) and γ (redundancy) over a 6×6 grid ($\alpha \in \{0, 0.25, 0.5, 0.75, 1.0, 1.5\}$, $\gamma \in \{0, 0.1, 0.25, 0.5, 0.75, 1.0\}$) on MTOP (ModernBERT-base, K=4): https://tinyurl.com/3acx74ry
>
> Key findings:
> - **Default:** α=1.0, γ=0.5 → 95.90% (near-optimal across the grid)
> - **Worst:** α=0.0, γ=0.0 → 94.80% (still +13pp over last-layer baseline: 81.37%)
> - **Total range:** only 1.35pp across the entire 6×6 grid
>
> Performance forms a broad, smooth plateau with degradation appearing only when both α and γ exceed 0.75, indicating over-constraining from simultaneous strong penalties rather than problematic interaction.
>
> We additionally validate hyperparameter robustness on ASVspoof LA19 (Wav2Vec 2.0, K=4), an audio domain unseen during hyperparameter selection:
>
>
> | Method     | α   | γ   | η    | K | ncal | Layers Selected | Test Acc ↑ |
> |------------|-----|-----|------|---|------|------------------|-------------|
> | Last Layer | -   | -   | -    | 1 | -    | -                | 0.90891     |
> | Last-4 Concat | - | - | - | 4| - | - | 94.83%
> | LOES       | 0   | 0   | 0    | 4 | 512  | 8, 11, 12, 10    | 0.95526     |
> | LOES       | 0   | 0.1 | 0    | 4 | 512  | 8, 0, 1, 2       | 0.97904     |
> | LOES       | 0   | 0.5 | 0    | 4 | 512  | 8, 0, 1, 2       | 0.97904     |
> | LOES       | 1   | 0.5 | 0.1  | 4 | 64   | 1, 0, 3, 2       | 0.98133     |
> | LOES       | 1   | 0.5 | 0.1  | 4 | 512  | 1, 0, 3, 2       | 0.98301     |
>
> LOES consistently outperforms all baselines across all hyperparameter settings, with less than 0.3pp variation, further confirming robustness across modalities. Table A2 further confirms that n_cal=64 (1% of data) is sufficient to achieve near-peak performance. The fixed default configuration works across all modalities without task-specific tuning, confirming the hyperparameters behave as decoupled regulators rather than interacting knobs.
>
> ## Q. Can you compute the expensive upper bound and show how far these heuristics are from this bound? Can you provide a scaling law for search compute vs final prediction accuracy?
>
> We exhaustively trained all $C(12,3)=220$ subsets for BERT-B (frozen), evaluating each on the test set:
>
> **MTOP (11 classes):**
>
> | | Layers | Test Acc |
> |---|---|---|
> | Global Optimal | [3, 6, 11] | 97.83% |
> | LOES | [11, 10, 6] | 96.67% |
> | Last Layer | [11] | 95.51% |
>
> **MTEB Emotion (4 classes):**
>
> | | Layers | Test Acc |
> |---|---|---|
> | Global Optimal | [3, 10, 11] | 71.08% |
> | LOES | [11, 10, 9] | 69.39% |
> | Last Layer | [11] | 68.40% |
>
> LOES achieves within 1.16pp (MTOP) and 1.69pp (Emotion) of the globally optimal subset, recovering the near-entirety of the exhaustive optimum's accuracy. Finding this global optimum required training and evaluating all 220 possible 3-layer combinations, a process that took several hours of GPU compute even for BERT-base (12 layers). LOES arrives at a competitive solution through a few closed-form calculations on a small calibration set followed by a single training run, requiring no combinatorial search whatsoever. We also observed that 2/3 layers were common highlighting that LOES was able to reach closely to the optimal solution.
>
> This computational gap grows rapidly with model depth. For ModernBERT (22 layers), exhaustive search would require $C(22,3)=1{,}540$ full training runs; for BERT-Large (24 layers), $C(24,3)=2{,}024$. These are precisely the architectures where LOES provides the largest downstream gains (Table 5), yet where exhaustive search is entirely infeasible. LOES offers a principled and scalable alternative that achieves near-optimal layer selection at a fraction of the cost.
>
> ### Scaling Law
> For scaling law analysis, we use MBERT-B on MTOP. Since exhaustive combinatorial search over layers is infeasible (for example, C(21,4) = 5985), we approximate an empirical upper bound by training layer-wise probes for 1, 3, and 5 epochs (Greedy1/3/5) and selecting the top-4 layers (Table 6 - https://tinyurl.com/ytm5awvp).
>
> These baselines exhibit a clear pattern of diminishing returns. Increasing the search budget substantially (168s to 840s, about 5x) yields only modest accuracy improvements (91.06 to 94.55, about 3.5 points), with gains saturating as compute increases.
> LOES matches this near-optimal performance (94.48 MTOP vs. 94.55) while requiring only 5.22s, over 160x less compute than Greedy5, demonstrating that it efficiently captures the most informative layers without exhaustive search.
>
> We will include this analysis in the revision.

---

> > ### Author Rebuttal · Reviewer_mPEw · 2026-04-03
> >
> > I remain very positive about this paper.

---

### Official Review · Reviewer_chK7 · 2026-03-12

**Soundness:** 2
**Presentation:** 2
**Significance:** 3
**Originality:** 3
**Overall Recommendation:** 4
**Confidence:** 3

**Summary:**

This paper investigates how deep intermediate representations in neural networks contain valuable information that is often ignored in standard transfer learning, which typically relies only on the final layer. The authors show that useful task-specific signals are distributed across multiple layers and propose LOES, a spectral method that identifies the optimal combination of intermediate layers to form a task-specific representation subspace. They also introduce GeoReg to maintain desirable representation geometry during adaptation. Experiments across multiple datasets demonstrate that selecting and combining intermediate layers consistently improves downstream performance, revealing that task-relevant information is not concentrated solely in the final layer but distributed across the network hierarchy.

**Compliance With Llm Reviewing Policy:**

Affirmed.

**Final Justification:**

Although there are some theoretical parts a bit hand-wavy (due to writing and assumptions) and need a more rigorous discussion, the paper tackles one important question and is a nice contribution to the field.

**Key Questions For Authors:**

1. To improve clarity, around line 112 “we therefore define …” It's better to illustrate with figures or examples to show how you use latent subspace as representation (given multiple layers, do you concatenate them or something else for layer fusion? Also related to line 211.) because this is quite different from standard representation learning settings where we only use the penultimate layer.
2. Experiments:
* Isn’t the performance improvement not surprising if you increase the feature dimension from just 1 last layer?
* (Important) Can you compare the performance with random layer selection and greedy selection (check linear probe accuracy for each layer and concatenate the max performance ones)?
* Does your selection method generalize (either theoretically or empirically) to non-linear probes?
* How much computational overhead is introduced with GeoReg (forward + backward)?
* (Important) I wonder if you can provide a more comprehensive Table 3? If my understanding is correct Tab.3 doesn’t include GeoReg loss. Therefore, I encourage authors to compare with more baselines (like random/greedy/last 3 layer concat). I think this is important because an effective and loss agnostic representation layer selection method can be widely applicable to many tasks. For one classification task, could you also plot TSNE/UMAP for the selected layers compared to some baseline?
* (Minor) In Table 4 can we include results for LOES without GeoReg?
* In Table 5 what are entropy, curvature and intrinsic?
* (Minor) In Figure 6, is the Urdu value missing in the radar plot for Last-3 Concat?
3. Theory:
* (Important) Motivation for LOES is only formally shown for the Isotropy term (heavily built on Assumption A.2’s spherical task prior though), but Red and Tri are somewhat arbitrary? For example, I imagine there will be other metrics that computes the spread of classes (maybe something related to uniformity, see Wang, Tongzhou, and Phillip Isola. "Understanding contrastive representation learning through alignment and uniformity on the hypersphere." International conference on machine learning. PMLR, 2020.). Why do you pick Tri? Red might be better illustrated with some figures because this is not too different from (ridge) linear regression?
(Important) Why is Geo-loss needed if (line 1423) the formulation aligns with the motivation of Iso? Then does this mean Geo-loss is redundant?
* Please reorganize the theoretical motivation A.6-A.9. There is too much content overlap with the main body without adding too much information. For example, A.7 is almost the same as Sec. 3.2.

**Limitations:**

Yes.

**Strengths And Weaknesses:**

## Strength
1. Significance & Originality: This is an important problem that may have a good amount of impact for many tasks. Also, the selection method is efficient. I was not aware of other principled representation layer selection methods. The authors at least tried to provide a theoretically grounded method here (but have flaws).

## Weakness
1. Soundness: some components of the LOES & Geo regularizer’s motivation is unclear. And some experiments are missing.
2. Presentation: There are a few clarity issues and the appendix (which I thought will have many useful contexts after reading the main paper) is poorly organized.

---

> ### Author Rebuttal · Authors · 2026-03-30
>
> We thank Reviewer chK7 for the detailed technical comments. Below, we address the weaknesses and questions with clarifications and supporting results. All changes below will be incorporated in the final revision.
>
> ## W1. Soundness
> We have updated Sec. 3.2–3.3 for the motivation of some components and have addressed most concerns in Q3.1 below.
>
> ## W2. Presentation
> We have reorganized the appendix with explicit cross-references between main text and appendix sections to improve navigation.
>
> ## Answers to Key Questions
> ### 1. Layer Fusion Mechanism
> Selected layers are combined via concatenation. For $K$ selected layers, the joint representation has dimension $\\sum_{i=1}^{K} d_{\\ell_i}$, which is fed to a linear classifier. We will include a figure illustrating this in the revision.
>
> ### 2. Experiments:
> 2.1 Controlling for Dimensionality
>
> - The **Last-k Concat** baselines directly controls for the dimensionality confound for vision tasks and audio tasks.
> - An example: Last-3 (Stanford Cars, DINOv2-S/14 Trainable) = 79.3% vs. LOES = **82.7%** (+3.4).
> - The same logic applies in the **language** domain, where **Last-4** baseline in **Table 5** provides an even cleaner comparison to LOES-4 across 7 datasets. For MBERT-B, Last-4 achieves 56.27% vs. LOES-4 at 64.89% (+8.62). For BERT-L, the gap is 3.22 points (70.11% vs. 73.33%).
>
>
> Since both methods use the same number of layers (same feature dimension), the gains are due to **which layers are selected**, not increased dimensionality.
>
> 2.2 Greedy and Random Layer Selection
> - We include **Random** baselines and a **Greedy** selection that picks the top 4 layers by probe accuracy (for fair comparison with LOES-4), evaluated on MBERT-B across two datasets. For Greedy we did training a probe per layer for 2 settings: 1 or 5 epochs, incurring notable overhead (~2.8 min for Greedy1, ~14 min for Greedy5).
> - LOES-4 outperforms Random (2-5), Greedy1 and matches Greedy5, while giving 32x and 160x faster layer selection than Greedy1 and Greedy5 respectively. (https://tinyurl.com/2ezcc7hf)
>
> 2.3 Non-Linear Probing
> We ran experiments with a ReLU non-linear probe. LOES generalizes to non-linear probes. Table: https://tinyurl.com/hyusjvh4
>
> 2.4 GeoReg adds ~5.6% per-batch overhead on DINOv2-S/14, Stanford Cars (trainable, K=3), dominated by eigenvalue decomposition of the D-dimensional covariance matrix (11ms for D=768). Table: https://tinyurl.com/mrytex48
>
> 2.5 Comprehensive Table 3 and t-SNE
>
> - We expand Table 3 for **DINOv2-S/14** (frozen and trainable), evaluating **Random-3**, **Last-3 layer concatenation**, and **learnable layer weighting** (all without GeoReg).  **Greedy** selection has been evaluated for text datasets (see 2.2 above).
> - LOES (k=3) achieves the best overall performance across most datasets, consistently outperforming both random selection and naive layer aggregation. Random-3 can be competitive on a few datasets but is highly inconsistent and significantly worse overall.
> - Table: https://tinyurl.com/yjjnufme
>
> - t-SNE on Stanford Cars (DINOv2-S/14, frozen) - https://tinyurl.com/4uvsm9rp
>
> 2.6 Results for LOES without GeoReg are present in Table 3.
>
> 2.7 Entropy, Curvature and Intrinsic In Table 5
> Entropy and Curvature are representation quality metrics from Skean et al. (2025) in paper L378-Right . Intrinsic refers to intrinsic dimensionality used to analyze intermediate LLM layers (Cheng et al., 2024; Razzhigaev et al., 2024) in paper L375-Right.
>
> 2.8 We thank the reviewer for spotting this error, yes the urdu value was missed while creating the plot. The value is there in the Table A7 though , which is **68.7** , we have updated the figure for the final paper.
>
> ### 3. Theory + W1:
> 3.1 We acknowledge that Section A.6 formally justifies only the **Isotropy** term ($\alpha$), while **Redundancy** ($\gamma$) and **Triangle Geometry** ($\eta$) are motivated empirically and by analogy.
>
> * **Redundancy (γ):** If $X_\ell \approx X_j$ for some $j \in S$, then $\\tilde{X}\ell \approx 0$, leading to an ill-conditioned ridge probe. $\mathrm{Red}\ell$ acts as an early warning, analogous to the $\ell_2$ term in Elastic Net vs. LASSO. We will formalize this in Appendix A.7.
>
> * **Triangle Geometry (η):** Captures the empirical link between centroid simplex volume and linear separability. No worst-case guarantees; $\eta=0$ for non-classification tasks (scope clarified in revision).
>
> - **GeoReg:** LOES enforces isotropy only at selection on frozen representations, whereas fine-tuning can distort this geometry. GeoReg maintains the same geometric conditions during training, ensuring the selected layers remain effective; accordingly, it has no effect when the encoder is frozen.
>
> 3.2 Reorganizing the theoretical motivation
> - We have refurbished the appendix organization for the final paper, reducing overlap with the main body.

---

> > ### Author Rebuttal · Reviewer_chK7 · 2026-04-03
> >
> > Thanks for the authors' very detailed rebuttal to all of my questions!
> > The additional experimental numbers seem impressive to me, but for the TSNE plot I actually didn't spot any immediate difference between 3 figures? Then what actually leads to the (subtle?) performance gain? I know sometimes TSNE can be unreliable, but for significant gains, usually we can see some better signs of linear separability. It would be nice to 1) actually include this figure in the paper 2) and give a VERY CONVINCING explanation here.
> >
> > -- After 2nd round of rebuttal ---
> > Thanks for authors' reply. My concerns are addressed and I will update my score accordingly.

---

> > > ### Author Response · Authors · 2026-04-04
> > >
> > > We thank the reviewer for acknowledging the additional experimental results. For the issues regarding t-SNE, we provide the following arguments:
> > >
> > > ### Explanation for Similar t-SNE Plots
> > > The similar appearance of frozen embeddings in t-SNE is expected. LOES ranks layers based on their linear accessibility of task-relevant information (via ridge regression; Eq. 1–3) and global geometric properties such as isotropy and redundancy. These are spectral, high-dimensional criteria, whereas t-SNE is a nonlinear 2D projection that may not faithfully preserve such distinctions. Consequently, improvements in downstream performance are not necessarily reflected in t-SNE plots of frozen features.
> > >
> > > We acknowledge that the t-SNE visualizations in the original submission were generated without training the probe and LayerNorm. We agree this limited the interpretability of the figures and thank the reviewer for highlighting this. In the last-3 and LOES-3 settings, the features are concatenated without layer normalization and probe, so the 3× higher dimensionality tends to dilute layer-specific differences in the 2D t-SNE projection, making the visual gap appear subtle compared to the last layer even when the downstream performance improves.
> > >
> > > To address this, we have now included t-SNE visualizations after training across multiple modalities (Figures 1–3), which provide a clearer and more convincing picture: https://docs.google.com/document/d/e/2PACX-1vSbcV1dmohxUM5k7BUuxRfw4PMvtwuC_eobxYz71LjZW4JGxyKJKg9Va8Ek98Lu3SR1AglI-IGQ0nWS/pub (Anonymous Link)
> > >
> > > Across all three modalities, LOES-selected layers produce tighter, more separated clusters compared to last-layer and last-K baselines, with performance gains being most pronounced for deeper models (ModernBERT, 22 layers).
> > >
> > > While frozen t-SNE plots may appear similar, the adapted representations (Figures 1–3) shows improved cluster compactness and separability, consistent with the gains in the paper. We will include these updated figures in the paper and provide in depth explanation for the same.
> > >
> > > Thank you for your thoughtful and constructive feedback it has significantly improved the quality and clarity of our paper. We hope that our revisions have adequately addressed your concerns, and we would be happy to provide any further clarification if needed.

---

### Official Review · Reviewer_rUkQ · 2026-03-12

**Soundness:** 3
**Presentation:** 2
**Significance:** 3
**Originality:** 2
**Overall Recommendation:** 4
**Confidence:** 2

**Summary:**

The paper studies the transfer learning capabilities of foundation models. In particular, it argues that using the final layer for transfer learning might not be universally optimal for all tasks. Based on this, a flexible layer selection framework is proposed where the task-optimal combinations of layers, under some geometry regularity constraints, are used to perform transfer learning. Experiments across multiple settings reveal that all layers, independent of their depth, can be relevant on a situational basis.

**Compliance With Llm Reviewing Policy:**

Affirmed.

**Final Justification:**

The paper provides a well engineered study demonstrating the relevance of intermediate layer representation in transfer learning. On the other hand, the non-unified diagnostic suite employed introduces an additional tuning overhead while it provides limited insights on the underlying mechanism allowing the increased transfer learning performance.

On balance, this is a solid paper where its weaknesses can be addressed by follow up works. The rebuttal reinforced my prior assessment.

**Key Questions For Authors:**

* Q1a. When fusing the selected layers, is there a dedicated transformation $W_l$ applied to the representation of each selected layer? Or the fine-tuning/transfer learning acts on the representation of the concatenated selected layers?
* Q1b. In relation to L188-left, why is the raw $X_{l*}$ used and not orthogonalized one? I found this to be counterintuitive given that the scores were computed on the orthogonalized representations.
* Q1c. How did you tune $\lambda$ is Eq. 2?

* Q2. What is the motivation behind the GeoReg formulation in Eq.9.? How does it compared with other structure preserving regularizations e.g. VICReg?

* Q3. It is argued in Sec 7. that the layer selection mechanism enables complementary view to existing probing methodologies. How does this relate to learned scalar combinations L103-right? (i.e., these could also, in principle, reveal insights on which layers are the most relevant for the task).

* Other points for your consideration:

    - L101-right is there a typo in the representation dimensionality (i.e.., $d$ to $d_l$)?
    - Consider updating the Algorithm 1 such that the pseudocode connects to the corresponding Equation (when applicable).
    - L152-154-right, this sentence is problematic given that it refers to Eq. 8 which is introduced later.
    - L159-162-right, is there any reference to that?

**Limitations:**

**Limitations:** Yes, in the supp material.

**Societal Impact:** No.

**Strengths And Weaknesses:**

**Strengths:**

* S1. The framing of combining multiple layers to increase fine-tuning/transfer learning performance is intuitive and well-motivated.
* S2. I found the connection between pretrained dataset diversity/scale and information distribution across layers very interesting.
* S3. The experiments are thorough and span across multiple pretrained encoders, datasets and modalities.

&nbsp;

**Weaknesses:**

* W1. Although the proposed framework achieves state-of-the-art performance, It introduces a number of tunable hyperparameters while doing so. While it is argued in L200-right that these hyperparameters are interpretable in isolation, their interaction is non-trivial to intuit.
* W2. I found the description of LOES, which is the key methodological contribution, to be unclear.
* W3. I found that the Geometric Regularization was not put into context with respect to previously explored techniques for avoiding representation collapse.

---

> ### Author Rebuttal · Authors · 2026-03-30
>
> We thank Reviewer rUkQ for detailed technical assessment. We address each point below. We will update the paper appropriately incorporating all these clarifications.
>
> ## W1: Many tunable hyperparameters
>
> The interpretability of hyperparameters can only be assessed in isolation and is shown in Appendix Table A1. To assess the interaction between them, refer to our 2D sensitivity analysis of α (isotropy) and γ (redundancy) over a 6×6 grid on MTOP (MBERT-B, K=4): https://tinyurl.com/3acx74ry
>
> Test accuracy forms a broad, smooth plateau. Our default (α=1.0, γ=0.5) is within 0.25pp of the best configuration (96.15%), and even the worst case remains +12-13pp over the last-layer and last4-concat baselines. Degradation occurs when both alpha and gamma exceed 0.75, suggesting that aggressive isotropy and redundancy penalization together can over-constrain the selection. Thus, hyperparameters behave as decoupled in most ranges. Performance is also robust for other modalities like audio as highlighted in the response to reviewer DEiU.
>
> ## W2: Description of LOES Unclear
>
> We acknowledge this concern. We have identified specific parts of Section 3.2 and Algorithm 1 for revision. The core intuition is: LOES greedily orthogonalizes candidate layer features against the current selection (Eq. 3), scores them via the composite objective (Eq. 7), and picks the best candidate. We will also add equation cross-references throughout Algorithm 1 to make the pseudocode-formalism connection unambiguous.
>
> ## W3: GeoReg Positioning
>
> We respectfully point the reviewer to Section 3.3, which already contextualizes GeoReg relative to VICReg and LeJEPA. We have now expanded this Section to explicitly highlight the key distinction: GeoReg is the only method of this family applied during fine-tuning on fused multi-layer representations with task-aware class-centroid separation.
>
> ## Q1a
>
> Each of the K selected layers passes through a LayerNorm followed by a linear projection to d=256. The K adapted representations are concatenated and fed to the classification head. LayerNorm aligns activation scales across layers so all contribute comparably while preserving feature geometry, following standard multi-layer fusion practice (Peters et al., 2018; Chiu et al., 2024).
>
> ## Q1b
>
> Orthogonalized features X̃_ℓ are used only during scoring to measure how much new information a candidate layer adds beyond the current selection. Once selected, the ridge probe is refit on raw features X_ℓ* against the original targets, since raw features retain full representational content including partially redundant directions that still improve ensemble calibration. Orthogonalization is used as a selection criterion & is not a representation transformation.
>
> ## Q1c
>
> λ was set to 1e-3 for numerical stability and not tuned per dataset. Upon the reviewer's prompting, we ran a sensitivity analysis on test accuracy for two MTEB tasks with ModernBERT-base:
>
> | λ | MTOP | AM-Scenario |
> |---|---|---|
> | Last-4 Concat | 79.23 | 63.35 |
> | 1e-5 | 94.80 | 79.02 |
> | 1e-4 | 94.27 | 79.62 |
> | 1e-3 | 94.45 | 78.45 |
> | 1e-2 | 94.73 | 79.76 |
>
> LOES outperforms Last-4 Concat by ~14-16 percent across all λ values, with less than 1.4% variation across the range.
>
> ## Q2
>
> GeoReg preserves spectral isotropy and class-centroid separation during fine-tuning where competing gradients can degrade them. On TweetEval-Emoji (BERT-base, K=3), without GeoReg validation accuracy degrades after ~5k steps due to representation collapse. Against a VICReg-adapted baseline, GeoReg achieves slightly higher performance (31.04% vs. 30.69%) with stronger collapse resistance, maintaining stable performance against the last-layer baseline (30.15%). A detailed comparison with VICReg, SIGReg, and additional regularizers will be added to the main paper; we omit here for character limits.
>
> ## Q3
>
> Learned scalar combinations (Peters et al., 2018) produce a weighted average over all layers, destructively blending features and providing only a soft ranking. In the transfer learning regime, scalar weights suffer from insufficient gradient signal in deep models, often converging to near-uniform distributions. LOES instead performs closed-form discrete selection on a small calibration set, concatenating a complementarity-preserving subset. Empirically, LOES k=3 outperforms the learnable weights baseline by 4.7pp on Stanford Cars (82.7% vs. 78.0%). Discrete selection is also more actionable: identifying that only K layers are task-discriminative enables pruning or distillation of the remaining layers, a benefit soft weighting cannot provide. This discussion will be added to the Appendix.
>
> ## Other Points
>
> We thank the reviewer for carefully spotting these issues. L101 typo ($d$ to $d_\ell$), Algorithm 1 equation cross-references, L152-154 reordering after Eq. 8, and L159-162 citation to Eq. 6 will all be corrected in the revision.
>
> We thank Reviewer rUkQ for the detailed and constructive feedback.

---

> > ### Author Rebuttal · Reviewer_rUkQ · 2026-04-02
> >
> > I would like to thank the authors for their response. I found that the responses addressed the majority of my concerns which I think will improve its presentation (W2) and the placement of the method in relation to previous works (W3). When it comes to W1, in my view, it is a fundamental limitation to the framework which limits its impact. Namely, multiple diagnostics are blended under a non-unified framing which, although shown to improve the performance, does not reveal the enabling factors mechanistically.
> >
> > On balance, my initial recommendation reflects my perception of the paper, and thus I maintain my score.
> >
> > Extra points for your consideration:
> >
> > - Tab 3 and 4 have the dataset (columns shuffled) unnecessarilly complicating the reading.
> > - Provide similar alpha/gamma ablations across multiple eta's to provide guidance for others using/extending your method.
> > - In the alpha/gamma ablation, the caption compares to the last layer baseline which, as also noted by chK7, is an unfrair comparison. Such unfrair comparisons can lead to wrong conclusions. For example one could infer that setting eta=0.1 (alpha=gamma=0) does the heavy lifting as it already boosts by ~12.5pp. However, to my understanding, part of the boost is to be atttributed to increased dimensionality.

---

> > > ### Author Response · Authors · 2026-04-04
> > >
> > > We thank the reviewer for maintaining the positive assessment and for the constructive suggestions. We address each point below.
> > >
> > > ## Dimensionality Clarification
> > >
> > > We appreciate the reviewer pointing out the unfair comparison with the last-layer baseline. We clarify that the appropriate baseline is the Last-K Concat, which uses the same number of layers (same fused dimensionality) as LOES and we use the same in our work. Last-4 Concat (Table 5) achieves 56.27% on MBERT-B vs LOES-4 at 64.89% (+8.62pp), and 70.11% on BERT-L vs 73.33% (+3.22pp), at identical dimensionality.
> > >
> > > ## Eta Ablation Across Multiple Values
> > >
> > > As requested, we provide alpha-gamma heatmaps at three eta values (0.0, 0.1, 0.2): https://ibb.co/XZMsVKQj
> > >
> > > The plateau shape is preserved across all eta values. Setting eta > 0 narrows the accuracy range from 2.10pp (eta=0.0) to 1.35pp (eta=0.1 and 0.2), confirming that the class geometry term acts as a stabilizer for the selection process rather than introducing additional sensitivity.
> > >
> > > We additionally sweep eta in {0, 0.05, 0.1, 0.15, 0.2, 0.3, 0.4, 0.5} at two configurations: https://tinyurl.com/44zzt7w6
> > >
> > > At the default configuration (alpha=1.0, gamma=0.5), the accuracy plateaus at 95.90% for eta in [0.05, 0.2] with only 0.91pp total variation across the full [0, 0.5] range. At alpha=1.0, gamma=0.25, the accuracy is completely flat at 95.90% from eta=0.05 through eta=0.5. The eta term exhibits threshold behavior: once it is strong enough to incorporate the final layer's class geometry into the selection (eta >= 0.05), further increases do not change the selected layers until eta >= 0.3, where over-constraining can degrade performance. We recommend eta in [0, 0.2] with 0.1 as a robust default.
> > >
> > > ## Table Column Ordering
> > >
> > > We thank the reviewer for noting the inconsistency in dataset column ordering between Tables 3 and 4. We will standardize the ordering in the revision.
> > >
> > > Thank you for your thoughtful and constructive feedback it has significantly improved the quality and clarity of our paper. We hope that our revisions have adequately addressed your concerns, and we would be happy to provide any further clarification if needed.

---

### Decision · Program_Chairs · 2026-04-30

**Decision:**

Accept (spotlight)

**Comment:**

This paper shows that task-relevant information in pretrained networks is distributed non-monotonically across layers and can be better exploited via principled selection and fusion rather than relying on the final layer output. To this end, the paper introduces a method for selecting a small subset of intermediate layers in deep models based on various geometric criteria (isotropy, redundancy, class structure) along with a regularization loss to preserve representation isotropic structure accordingly during fine-tuning.

Reviewers agree the paper addresses an important and practical problem and provides a technically sound solution with strong empirical support. The breadth of evaluation across vision, language, and speech, and consistent improvements over last-layer and standard fusion baselines, are key strengths.
Concerns included: clarity; hyperparameter sensitivity and limited ablations, and whether gains come from selection vs. increased capacity. Based on the rebuttal, these are largely addressed: the method and fusion pipeline are clarified, stronger baselines and random/greedy comparisons are added, exhaustive search is approximated, and robustness is better supported. Several reviewers indicated their concerns were resolved.

Therefore, I recommend strong acceptance. The work is technically sound, well-evaluated, and likely to be useful for representation learning and transfer with pretrained models, therefore interesting for multiple sub-communities attending ICML. Remaining issues are primarily about presentation and positioning.

For the final version, the authors should:
- improve clarity of LOES and link equations to the algorithm
- clearly separate theoretical vs. empirical motivations
- integrate key rebuttal results (stronger baselines, exhaustive search comparison, corrected tables).